# Regulation of cerebral blood flow boosts precise brain targeting of vinpocetine-derived ionizable-lipidoid nanoparticles

Xufei Bian[1,2], Ling Yang[1], Dingxi Jiang[1], Adam J. Grippin [3], Yifan Ma[3], Shuang Wu[1], Linchong Wu[1], Xiaoyou Wang[1,2], Zhongjie Tang[1], Kaicheng Tang[1], Weidong Pan[4], Shiyan Dong [3], Betty Y. S. Kim [5], Wen Jiang [3] ✉, Zhaogang Yang [6] ✉ & Chong Li [1,2] ✉

Despite advances in active drug targeting for blood-brain barrier penetration, two key challenges persist: first, attachment of a targeting ligand to the drug or drug carrier does not enhance its brain biodistribution; and second, many brain diseases are intricately linked to microcirculation disorders that significantly impede drug accumulation within brain lesions even after they cross the barrier. Inspired by the neuroprotective properties of vinpocetine, which regulates cerebral blood flow, we propose a molecular library design centered on this class of cyclic tertiary amine compounds and develop a self-enhanced brain-targeted nucleic acid delivery system. Our findings reveal that: (i) vinpocetine-derived ionizable-lipidoid nanoparticles efficiently breach the blood-brain barrier; (ii) they have high gene-loading capacity, facilitating endosomal escape and intracellular transport; (iii) their administration is safe with minimal immunogenicity even with prolonged use; and (iv) they have potent pharmacologic brain-protective activity and may synergize with treatments for brain disorders as demonstrated in male APP/PS1 mice.

As the world population ages, the incidence of brain-related diseases has risen significantly, and such diseases are now the leading cause of mortality[1]. A pressing challenge lies in the dearth or absence of disease-modifying drugs for patients afflicted with these debilitating brain disorders, despite substantial and continuous investments. This is most apparent in the case of Alzheimer disease (AD), for which the overall success rate in drug development stands at a mere 0.5% of 1120 distinct pipeline drugs, representing the lowest rate among all therapeutic areas[2]. Moreover, the presence of the blood-brain barrier (BBB) blocks the application of numerous cutting-edge therapies aimed at treating central nervous system disorders[3]. Nucleic acid-based biopharmaceuticals, for instance, face challenges due to their limited brain penetration, hindering their potential for traditionally perceived "undruggable" or "untreatable" targets in the brain[4]. Although active targeting strategies based on receptor-mediated transcytosis have been used to enhance BBB permeability[5] and some progress has been made in their clinical application (e.g., ANG1005[6], 2B3-101[7], and JR-141[8]), at least two major factors remain that impede efficient brain-targeted delivery that have not been adequately explored or addressed. First, the attachment of a targeting ligand to the drug or drug carrier does not enhance its biodistribution within the brain; rather, it primarily improves cellular internalization only when encountering the targeted cells but can still have limited BBB penetration[9]. A critical yet

[1]Medical Research Institute, College of Pharmaceutical Sciences, Southwest University, Chongqing, PR China. [2]State Key Laboratory of Natural and Biomimetic Drugs, Peking University, Beijing, PR China. [3]Department of Radiation Oncology, The University of Texas MD Anderson Cancer Center, Houston, TX, USA. [4]School of Pharmaceutical Sciences, Guizhou University, Guiyang, PR China. [5]Department of Neurosurgery, The University of Texas MD Anderson Cancer Center, Houston, TX, USA. [6]School of Life Sciences, Jilin University, Changchun, PR China. ✉e-mail: wjiang4@mdanderson.org; zhaogangyang@jlu.edu.cn; chongli@swu.edu.cn

often overlooked aspect is that many brain diseases are linked with small cerebrovascular embolisms and microcirculation disorders[10] that significantly impede drug accumulation and therapeutic efficacy within lesions even after the drug crosses the BBB. Examples include the progression from mild cognitive impairment to AD being accompanied by a concurrent global and severe reduction (up to 40%) in cerebral blood flow[11]; meningitis-induced inflammation involving intracranial vessels, leading to thrombosis and subsequent ischemia[12]; and brain tumors inducing stroke through direct compression of blood vessels[13]. At present, assessments of active brain-targeting predominantly rely on healthy animal models or in vitro models under physiological conditions, models that fail to fully capture the adverse effects of pathological reduction of cerebral blood flow on drug delivery to the site of lesions.

Vinpocetine, a compound derived from the indole alkaloid vincamine found in *Vinca minor* leaves, has been used extensively as a prescription medication or dietary supplement for cerebrovascular disorders and cognitive impairments such as stroke, senile dementia, and memory disorders[14,15]. Its beneficial effects are purportedly attributable to its selective inhibition of brain phosphodiesterase type 1 (PDE1)[16], which contributes to the maintenance or restoration of physiological cerebral vascular dilation, recovery of normal cerebral blood flow in ischemic areas, and selective augmentation of regional cerebral blood flow without significantly affecting systemic blood pressure[17]. In essence, the distinctive capability of vinpocetine to selectively enhance cerebral blood flow presents an opportunity for optimizing brain-targeted drug delivery formulations. Currently, substantial research involving quantitative structure-activity relationship analysis has made noteworthy progress in elucidating the structural characteristics and functional aspects of the vinpocetine family of indole alkaloids. These advancements have laid a solid foundation for recognizing their skeletal structures and functions, thereby facilitating drug research and development. Currently four drugs from this family are commercially available, five compounds are in clinical trials, and numerous other compounds are in preclinical investigations[18–21]. In terms of structural attributes, the indole-fused heterocycles present in vinpocetine possess a "protonable" amine structure. Consequently, they should exhibit predictable electrostatic interactions with nucleic acid drugs and have potential as core scaffolds for designing carrier materials capable of loading nucleic acid drugs or other therapeutic agents.

Taking inspiration from the regulation of cerebral blood flow and the structural characteristics of vinpocetine, we have developed an innovative brain-targeting delivery system termed vinpocetine-derived ionizable-lipidoid nanoparticles (VIP). This ionizable-lipidoid-based system has been shown to have at least four distinct effects and to successfully deliver various types of cargo, including siRNA, mRNA, and small-molecule drugs, for the treatment of brain diseases such as AD, brain tumors, and infections. Notable aspects of this VIP delivery system are as follows: (i) It achieves enhanced brain targeting through a mechanism involving selective regulation of cerebral blood flow, distinguishing it from existing active brain-targeting strategies. (ii) It demonstrates efficient loading and delivery capacity, in particular facilitating the escape of nucleic acid drugs from endosomes and improving intracellular transport. (iii) Relative to the gold standard ionizable lipid DLin-MC3-DMA, which is clinically approved for RNA therapies[22], the VIP system produces no significant inflammation at the single-cell level and significantly reduces the incidence of hepatorenal toxicity or complement activation-related pseudoallergy (CARPA)[23] during long-term administration. (iv) The vinpocetine derivatives used in the VIP system retain their inherent pharmacologic functions and can serve as synergistic therapy adjuvants[24], which may be useful in light of the complex nature of brain diseases, which are often accompanied by cerebrovascular lesions[25].

# Results

## Design and screening of VIP lipidoids
Based on the structures of vincamine, vinpocetine, and their synthetic intermediates[26], a dozen amine head groups were preliminarily screened, and representative five-, four-, and three-membered rings were selected (Fig. 1a). The tail chains have diverse, representative structural characteristics with unsaturated bonds, ester groups, and alcoholic hydroxyl groups that facilitate gene delivery[27] (Fig. 1b). Helper lipids (DSPC, DOPC, DOPE) and DMG-PEG$_{2000}$, together with cholesterol (Chol) and proposed ionizable lipidoids, were used for nucleic acid delivery. The relative molar percentages of lipids in VIP were designed as follows: key lipid (40–60%), helper lipid (10–15%), Chol (23–49%), and DMG-PEG$_{2000}$ (1–2%) (Supplementary Table S1; the sum of molar ratios of the four lipids was 100 for each formulation).

We conducted multiple screening rounds for ionizable lipid molecules by using blank lipid nanoparticles (LNPs) (Fig. 1c). First, we chose optimal formulation 14 based on the size of the preparations (Supplementary Table S2) and went on to prepare VIP with 18 candidate compounds that were smaller than 100 nm (Supplementary Table S3). Next, we measured the acid dissociation constant (pKa) of nine eligible compounds and found the values to be within the range required for mRNA delivery (6.2–6.5) (Supplementary Fig. S1). Because vinpocetine selectively inhibits $Ca^{2+}$-calmodulin-dependent cGMP-phosphodiesterase to enhance intracellular cGMP levels in the vascular smooth muscle (leading to reduced resistance in cerebral microvessels and increased cerebral microvessel blood flow)[28], the third screening focused on whether these derivatives retained this property. At 10 min after injection, a faster and greater increase in cerebral blood flow was observed in the groups treated with VIP prepared with A1-B1-C3.3 ($121.8 \pm 10.4\%$ of the baseline value), A3-B1-C3.2 ($129.7 \pm 8.4\%$), and A5-B1-C4.2 ($130.4 \pm 7.6\%$) relative to vinpocetine ($112.0 \pm 4.9\%$), and that increase persisted for 60 min (Fig. 1d–f). Interactions between the three vinpocetine-derived compounds and PDE1 were measured by using a surface plasmon resonance binding assay. The kinetic parameters of the interactions (Fig. 1g and Supplementary Table S4) were consistent with those of the molecular docking simulation (Fig. 1h). Overall, compound A5-B1-C4.2 had the highest selectivity; its affinity for PDE1 reached 62.9 nM; and it exhibited no specific interactions with albumin.

Finally, the representative ionizable lipids (A5-B1-C4.2) were generated and synthesized as described in Methods in the Supplementary Materials. The compound structures were confirmed by [1]H nuclear magnetic resonance (NMR) and [13]C NMR (Supplementary Fig. S2).

## Preparation and characterization of siRNA-loaded VIP
The leading structure (A5-B1-C4.2) and formulation composition (molar ratio A5-B1-C4.2:DOPC:cholesterol:DMG-PEG = 50:12.5:36:1.5) were screened and selected. The encapsulation efficiency was $91.1 \pm 2.77\%$, and the zeta potential of VIP@siRNA decreased sharply to nearly 0 mV at neutral pH. The VIP@siRNA was round and had a hydrodynamic size of approximately 90 nm (Fig. 2a, b). The siRNA was completely entrapped by VIP when the mass ratio was higher than 1:2, based on a gel retardation assay (Fig. 2c).

To investigate the transfection behavior of the VIP, we encapsulated FAM-labeled siRNA into the VIP, with NP as a negative control and MC3, a commercially available LNP, for comparison (Supplementary Table S5). The results showed that the VIP had high transfection and endosomal escape efficiencies. In addition, the intracellular mean fluorescence intensity (MFI) was 3.75 times that of the commercial DLin-MC3-DMA (Fig. 2d), and the colocalization of siRNA and endosomes/lysosomes was equivalent to that of the MC3 group (Fig. 2e). Next, we investigated the endocytic pathway. Compared with VIP@FAM-siRNA transfection without an inhibitor, the transfection efficiency of cells pretreated with different inhibitors varied significantly, and filipin and chlorpromazine significantly

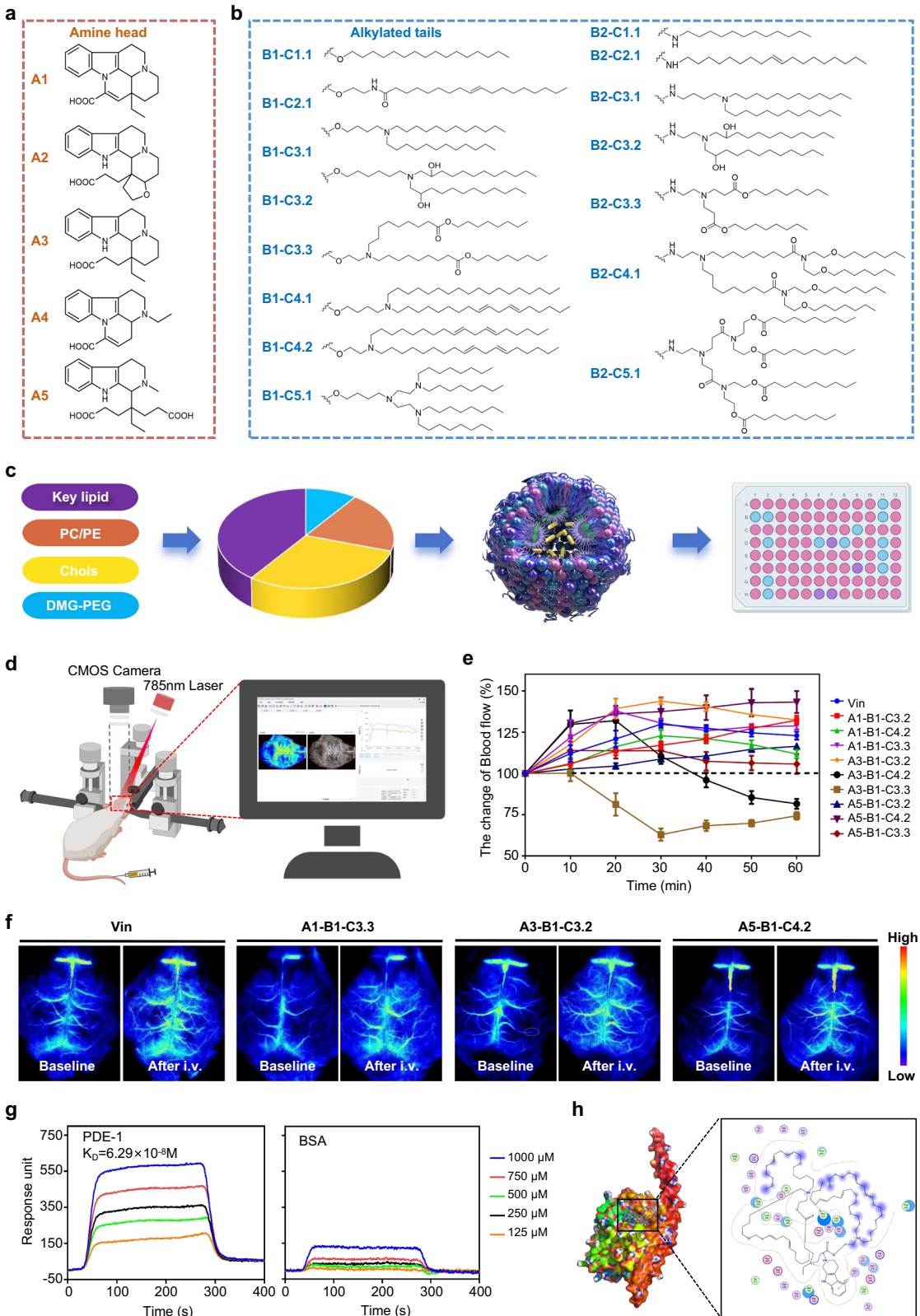

**Fig. 1 | Screening and component optimization of the ionizable-lipidoid molecule based on the skeleton of vinca alkaloids (vinpocetine). a, b** Chemical structures of lipid building blocks, including 5 amine heads and 15 alkylated tails. **c** Optimization scheme for VIP; 27 formulations were prepared and evaluated in this study based on 75 ionizable compounds. (Created with BioRender.com). **d** Schematic diagram of laser speckle flowmetry for detecting cerebral micro-circulation blood perfusion in mice. (Created with BioRender.com). **e** Quantitative curve of blood flow changes of 9 candidate compounds and vinpocetine (Vin) within 60 min after administration. Data are presented as means ± SD, were obtained by measuring the blood perfusion unit every 10 min. **f** Laser speckle flowgraphy images of 3 candidate compounds and vinpocetine (Vin). **g** Molecular interaction between the lipidoid molecule (A5-B1-C4.2) and PDE1 protein investigated through surface plasmon resonance (SPR). **h** Three-dimensional ligand-protein interaction mode for the binding site of PDE1 (Uniport ID: Q14123) with the leading compound A5-B1-C4.2. Data in (**e**, **f**) are representative of two independent experiments with similar results. Source data are provided as a Source Data file.

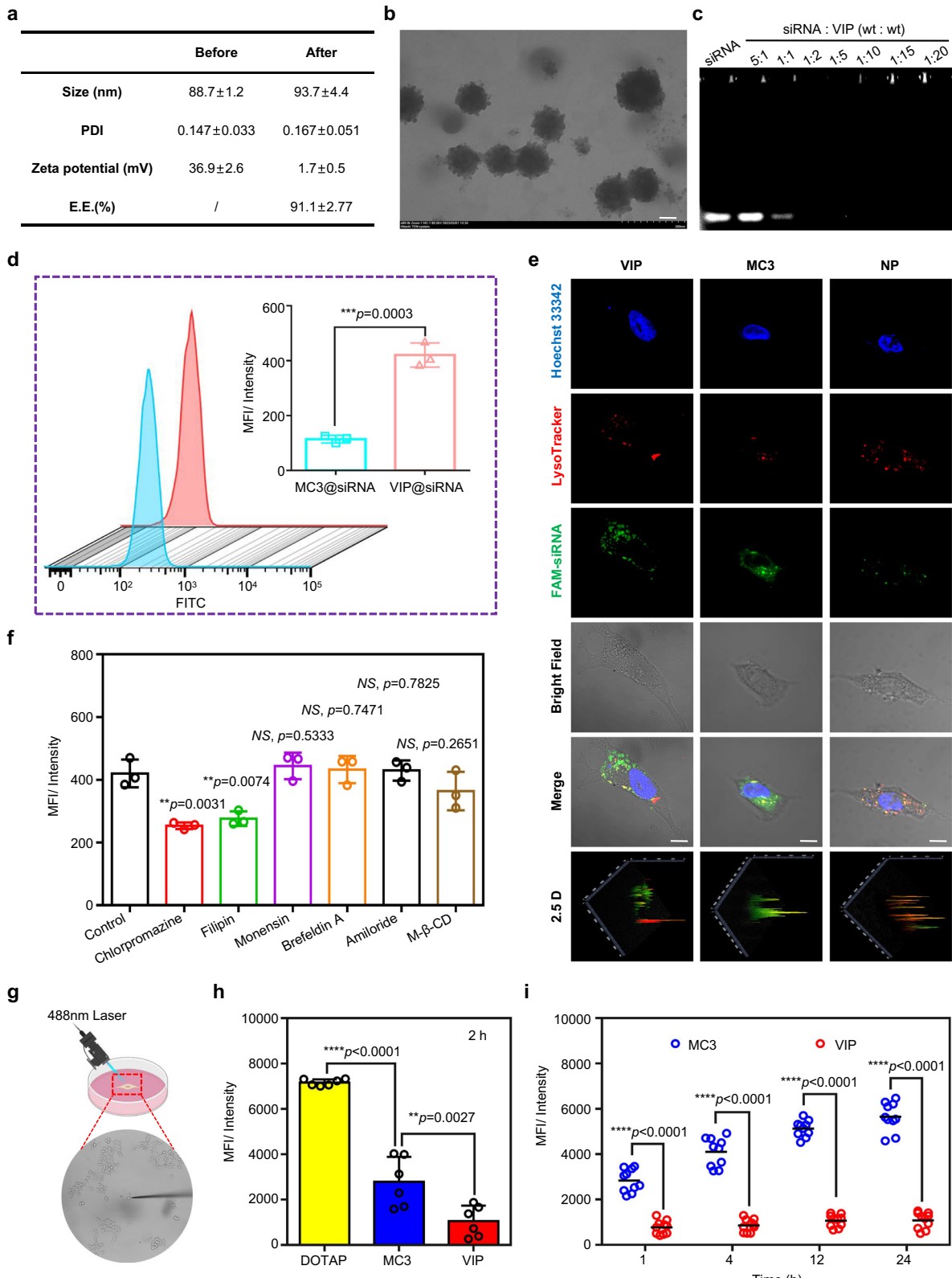

repressed the cellular entry of VIP@siRNA mediated by caveolae and clathrin[29] (Fig. 2f).

The stability of A5-B1-C4.2 and VIP@siRNA was further evaluated as follows. A5-B1-C4.2 did not degrade at 40 °C, at room temperature (approximately 20 °C), or at −20 °C for at least 2 weeks (Supplementary Fig. S3), and VIP@siRNA remained stable for at least 2 weeks at 4 °C (Supplementary Fig. S4). Moreover, VIP@siRNA had good stability in phosphate-buffered saline (PBS) and 10% fetal bovine serum (FBS) (Supplementary Fig. S5), which is crucial for the development of RNA interference (or mRNA) therapies to be stored and transported at refrigerated or room temperatures. The potential cytotoxicity of ionizable lipids has raised broad concerns[30], and discontinuous measurements and end-point data collection do not reflect time-varying, dynamic changes. Also, these traditional toxicity detection methods

**Fig. 2 | Biophysical characterization and in vitro studies of VIP@siRNA.**
**a** Physicochemical properties of VIP before and after siRNA encapsulation, including size, polydispersity index (PDI), zeta potential, and encapsulation efficiency (E.E.). **b** Transmission electron microscopy image of VIP@siRNA; scale bar = 50 nm. **c** Gel retardation assay of VIP@siRNA at siRNA/LNP weight ratios of 0.2, 1, 2, 5, 10, 15, and 20. **d** Detection of cellular transfection by flow cytometry and comparison of mean fluorescence intensity of VIP@siRNA and MC3@siRNA. Data are presented as means ± SD (n = 3 biologically independent samples). **e** Subcellular localization of VIP@siRNA or MC3@siRNA; scale bar = 5 μm. **f** Exploration of mechanisms underlying VIP-mediated siRNA transfection. Data are presented as means ± SD (n = 3 biologically independent samples). **g** Schematic diagram of a real-time single-cell multimodal analyzer combined with a fluorescent probe to detect reactive oxygen species levels in a single-cell. (Created with BioRender.com). **h** Reactive oxygen species production in bEnd.3 cells after administration of high doses (0.25 mg/mL) for 2 h. Data are presented as means ± SD (n = 6 biologically independent samples). **i** Reactive oxygen species production in bEnd.3 cells after administration of low doses (0.05 mg/mL) for 1 h, 4 h, 12 h, 24 h. Data are presented as means ± SD (n = 10 biologically independent samples). $*P < 0.05$, $**P < 0.01$, $***P < 0.001$, $****P < 0.0001$, NS means no significance. Statistical significance was calculated with two-tailed unpaired t-tests. Data are representative of three (**b**, **c**) and two (**e**) independent experiments with similar results. Source data are provided as a Source Data file.

rely on macro- or micro-sized probes that are incompatible with the localized detection of indicators related to cytotoxicity. Previous studies[31] have confirmed the presence of spatial heterogeneities in biomolecule distribution and biochemical processes (e.g., oxidative stress) in single cells. Therefore, we used a real-time quantitative single-cell detection method to investigate oxidative stress in living cells. Reactive oxygen species generation was found to be induced by VIP at much lower dose levels over time compared with commercial DOTAP and MC3 (Fig. 2g–i), which is consistent with routine cytotoxicity tests (Supplementary Fig. S6).

### In vitro and in vivo brain targeting of VIP
Our in vitro quantitative flow cytometry analysis revealed increased uptake of VIP (2.56× that of MC3) (Fig. 3a), which led us to test the effects of downregulating the expression of PDE1 protein with siRNA. At 72 h after siPDE1 transfection, protein expression was reduced by approximately 50% compared with the negative control (Fig. 3b). The decreased uptake of VIP in the cell model with lower PDE1 expression suggested that VIP may have been interacting with the PDE1 adjacent to the cell surface (Fig. 3c). Because vinpocetine can participate in reversal of drug efflux by inhibiting p-gp[32], we confirmed that VIP had a similar effect, which may also have contributed to the accumulation of fluorescent dye or drugs in the cells (Supplementary Fig. S7a). Indeed, a Transwell model demonstrated that penetration of control nanoparticles (NP) and MC3 was significantly lower in the VIP-treated condition (Fig. 3d, Supplementary Fig. S7b and Table S5). Quantitative analysis showed increased penetration through the BBB membrane model; the VIP-treated group had 3.75× as much nanoparticles in the basal compartment than the proportion in the MC3 condition (Fig. 3e). These results indicated that VIP could cross the BBB and be subjected to endocytosis by endothelial cells. We next labeled NP, VIP-siNC (siNC=siNegative control), and MC3-siNC with the fluorescent dye DiD and measured plasma levels of DiD after intravenous injection of these compounds to evaluate their in vivo pharmacokinetics. We found that VIP and MC3 had similar blood circulation times and elimination half-lives (t$_{1/2}$) (Fig. 3f), and we found that combining the A5-B1-C4.2 with LNP (i.e., VIP) prolonged its in vivo residence time relative to A5-B1-C4.2 alone (Supplementary Fig. S8). To evaluate the brain-targeting efficiency of nanoparticles in vivo, we injected different LNPs labeled with DiD into normal mice with head hair removed and monitored them with live fluorescent imaging (Fig. 3g). The nanoparticles with VIP showed significantly stronger intensity in the brain than those without VIP, validating their potent brain-targeting capacity. At 0.5 h, 1 h, 2 h, 4 h, and 8 h after injection, major organs were removed from the mice for ex vivo imaging (Fig. 3h and Supplementary Fig. S9). Mice treated with VIP showed stronger fluorescence signals in the brain than the positive control, GSH-LNP (in which glutathione [GSH] was used as the target moiety, Supplementary Table S5); a GSH-modified liposomal formulation is currently in clinical trials for brain-targeted delivery[7]. Quantitative results further showed that the fluorescence intensity of the brain in the VIP group was approximately 209% that in the positive control group (Supplementary Fig. S9). We also tested the effects of pretreating mice with four different agents known to reduce cerebral

blood flow (5-hydroxytryptamine, dexmedetomidine, $N^G$-mono-methyl-L-arginine, and noradrenaline) and found that all of these agents significantly reduced cerebral blood flow and reduced the accumulation of VIP-loaded fluorescent dye in the brain tissues, indicating that regulation of cerebral blood flow has a central role in targeting VIP to the brain (Supplementary Fig. S10 and Table S6).

We also demonstrated targeting of the VIP delivery system in three classical models of brain disease, an approach that differs from many studies that examined brain targeting only in normal mice. One of these models, APP/PS1 mice[33], show accelerated amyloid deposition and synaptic loss with reliable memory deficits, similar to a typical AD model; cerebral gliomas[13] and fungal meningitis[12] are known as being high risk with low cure rates. We verified experimentally that the three representative models all had cerebral microvascular blockage and decreased local cerebral blood flow[10,12,34], which could restrict drug accumulation. As the disease progressed, the cerebrovascular lesions and ischemia became increasingly severe, and the cerebral microvascular distribution was difficult to observe. Nevertheless, VIP was found to improve cerebral blood flow after a single injection regardless of the disease stage. In conclusion, we found that VIP improved cerebral blood flow in normal mice and also enhanced cerebral blood flow at the lesion site in these three pathological models at different stages (Fig. 3i and Supplementary Fig. S11), thereby enhancing brain targeting. Notably, pharmacologic modulation of cerebral blood flow has also been shown to have significant positive effects on the BBB under pathological conditions[35,36], suggesting that the development of VIP-based strategies may have multidimensional (potentially additive) therapeutic effects.

### Brain-targeted delivery of siBACE1 by VIP for the comprehensive treatment of AD
First, we evaluated the performance of the delivery system by using AD as an evaluation model and beta-site amyloid precursor protein-cleaving enzyme 1 (BACE1)[37] as a target to explore whether the systemic delivery of siRNA could elicit pharmacologic effects. After preliminary experiments using female and male APP/PS1 mice, we determined that there was no discernible gender bias concerning cerebral blood flow enhancement and BACE1 protein silencing in the APP/PS1 mice model treated by the VIP system (Supplementary Fig. S12).

Male APP/PS1 mice were given VIP@siBACE1 or control VIP@siScr (siRNA, 1 mg/kg) via caudal vein injection every 2 d for 2 weeks (Fig. 4a). The same dose of MC3@siBACE1 was used as another control. PBS-injected APP/PS1 and control wild-type (WT) mice were included to ascertain AD-relevant deficits in APP/PS1 mice at baseline. At the end of these treatments, spatial learning, and memory were assessed with the novel object recognition (NOR)[38] and then Morris water maze (MWM)[39] tests.

In the NOR test (Fig. 4b), after treatment with VIP@siBACE1, APP/PS1 mice showed a significant increase in NOR compared with PBS-treated APP/PS1 control mice, and the discrimination index (DI) and preference index (PI) for the novel object reached the performance levels of normal WT mice (Fig. 4c, d). Results on the MWM test showed

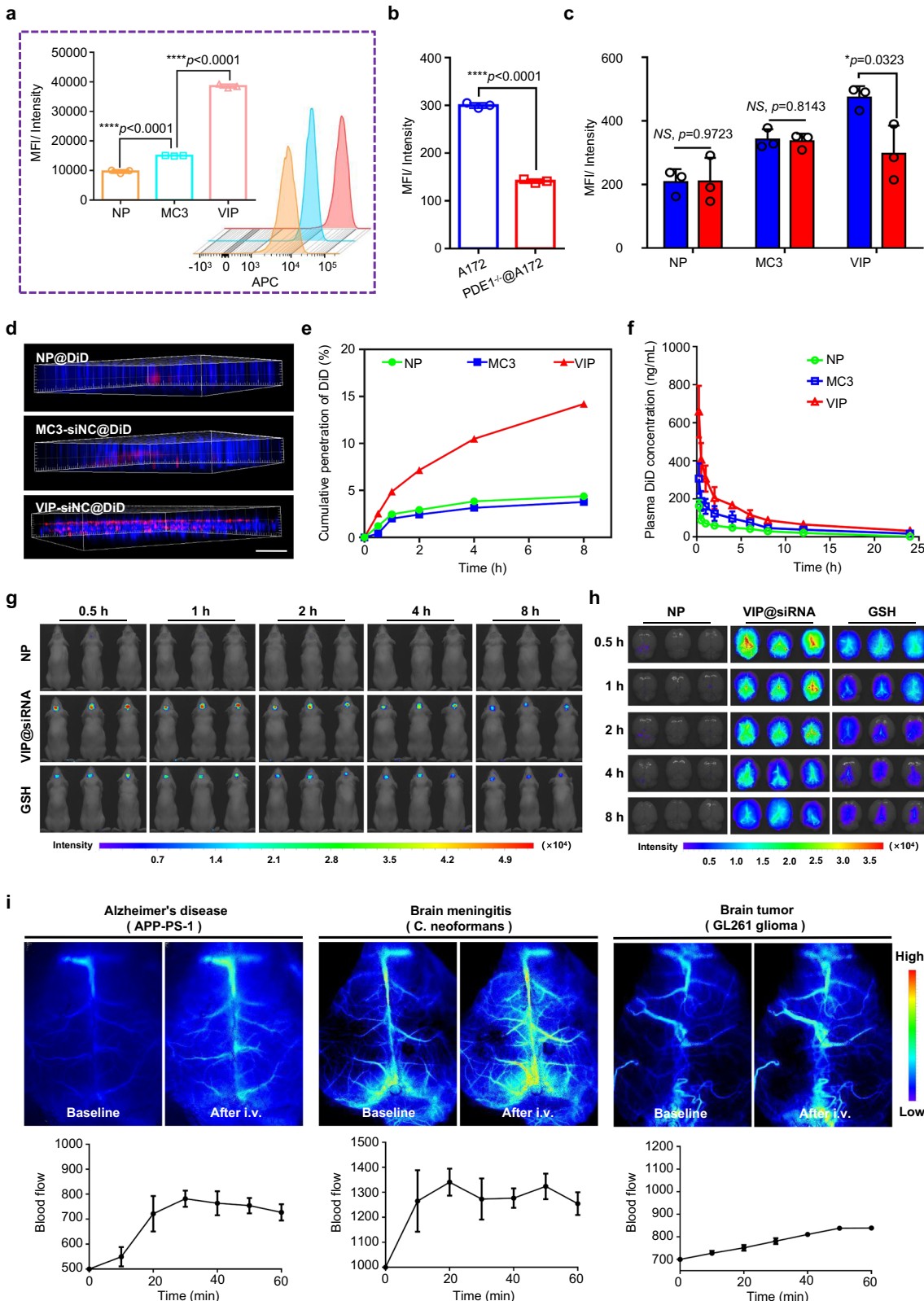

a similar trend: APP/PS1 mice treated with VIP@siBACE1 demonstrated a significant decrease in escape latency during the first 5 days of training relative to the PBS group and revealed a learning pattern similar to that of the WT group (Fig. 4e). In contrast, mice given PBS and MC3@siBACE1 showed an aimless search strategy with no improvement in spatial learning or memory (representative tracking plots are shown in Fig. 4f), and no difference was noted in swimming speed (Fig. 4g). VIP@siBACE1-treated APP/PS1 mice showed the greatest improvement, with a 2.19-fold increase in the percentage of time spent in the target quadrant and the largest number of crossings compared with PBS-treated APP/PS1 control mice (Fig. 4h, i). These data confirm that VIP@siBACE1 mediates highly effective siRNA delivery to significantly improve cognitive performance in APP/PS1 mice.

**Fig. 3 | Validation of brain targeting and related mechanism. a** Flow cytometry detection of cellular uptake of LNP. Data are presented as means ± SD (n = 3 biologically independent samples). **b, c** Flow cytometry detection of downregulated expression of PDE1 with siRNA and corresponding cellular uptake. Data are presented as means ± SD (*n* = 3 biologically independent samples). **d, e** An in vitro blood-brain barrier (BBB) model was established with a Transwell assay to verify the permeability of LNP through the barrier; scale bar = 1000 μm. **f** Plasma DiD concentration in SD rats after intravenous administration of NP@DiD, MC3-siNC@DiD, or VIP-siNC@DiD. Data are presented as means ± SD (*n* = 3 biologically independent

samples). **g, h** In vivo imaging of mice at 0.5 h, 1 h, 2 h, 4 h, and 8 h after administration of LNP. **i** Blood flow enhancement effect of VIP on brain microvessels detected by using laser speckle flowgraphy in three classical models of brain disease. Data are presented as means ± SD, were obtained by measuring the blood perfusion unit every 10 min. *P < 0.05, **P < 0.01, ***P < 0.001, ****P < 0.0001, *NS* means no significance. Statistical significance was calculated with **a, b** two-tailed unpaired *t*-tests and **c** multiple *t*-tests. Data in (**d, i**) are representative of two independent experiments with similar results. Source data are provided as a Source Data file.

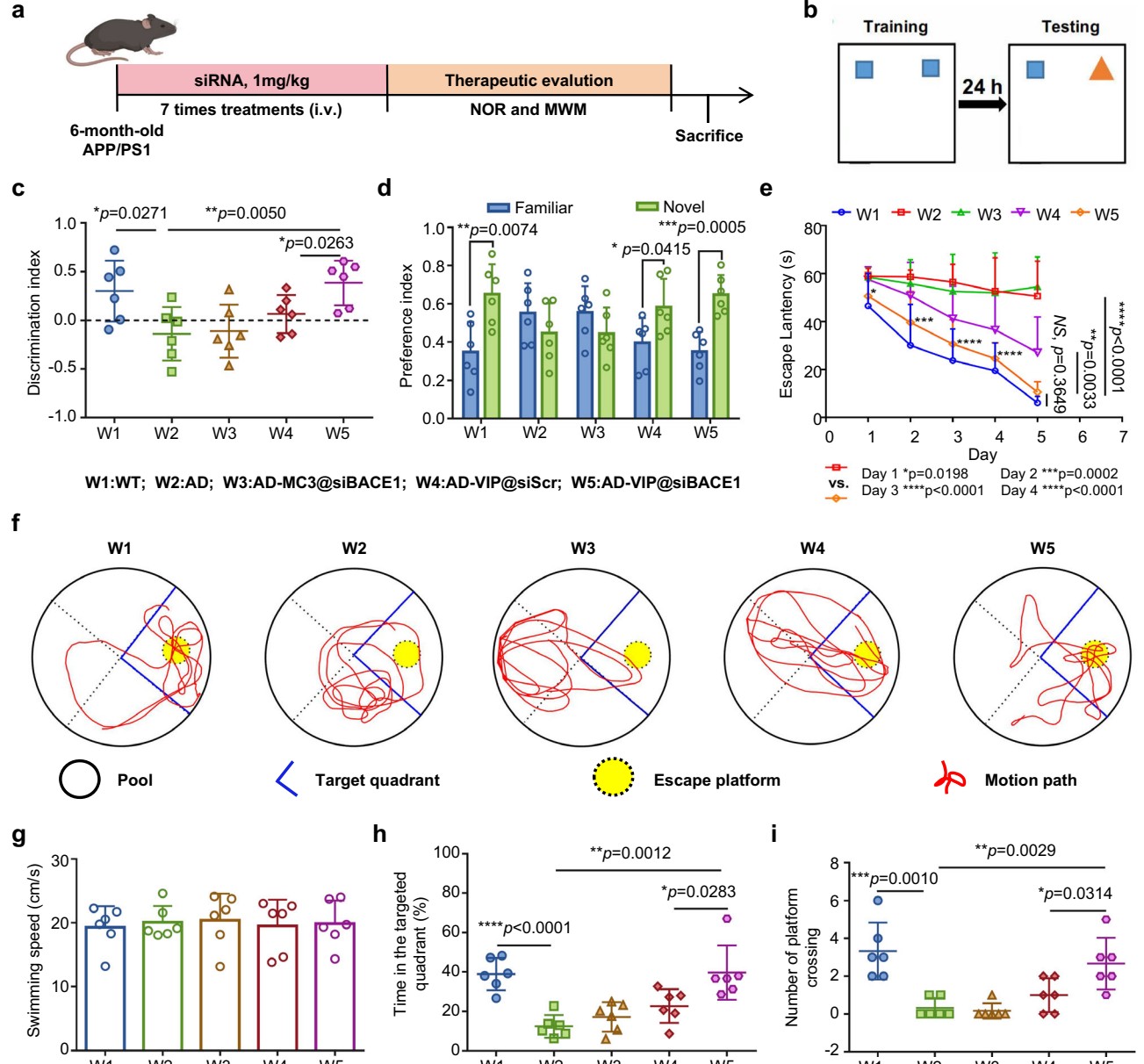

**Fig. 4 | Behavioral evaluation of VIP@siBACE1 therapy in APP/PS1 mice.**
**a** Schematic of the experimental timeline; APP/PS1 and wild-type (WT) mice were treated with LNP@siBACE1 or PBS via tail vein injection every 2 d (7 cycles). Mice were then subjected to the novel object recognition (NOR) and Morris water maze (MWM) tests to evaluate memory, and samples were collected for molecular pathological assessments. (Created with BioRender.com). **b** Setup for the NOR test. **c, d** Results from the NOR test. **c** Discrimination index and **d** preference index of each group after LNP@siBACE1 treatment. **e–i** Data in the MWM. **e** The 5-day

learning curve for the MWM experiment. **f** Representative swimming track, **g** swimming speed, **h** ratio of time spent in the target quadrant, and **i** numbers of crossing the platform location of each group on the probe test day. Data are presented as means ± SD (*n* = 6 biologically independent samples). *P < 0.05, **P < 0.01, ***P < 0.001, ****P < 0.0001. Statistical significance was calculated with **c, e, h, i** two-tailed unpaired *t*-tests and **d** multiple *t*-tests. Source data are provided as a Source Data file.

After the behavioral tests were completed, the mice were euthanized, and brain tissue was collected to analyze BACE1 suppression and its effects on Aβ and tau pathological accumulation (Fig. 5a–f and Supplementary Figs. S13, S14). Our results showed that hippocampal and cortical BACE1 protein levels in the VIP@siBACE1-treated APP/PS1 mice were significantly lower than those in the other APP/PS1 control groups (Fig. 5a and Supplementary Fig. S13a). The pathological hallmark of AD, amyloid plaques[40,41] derived from BACE1-cleaved APP, were significantly decreased in number and size of foci in the hippocampus and cortex of VIP@siBACE1-treated APP/PS1 mice (Fig. 5d, e). Another major pathological feature of late-stage AD is the development of intracellular neurofibrillary tangles composed of hyperphosphorylated tau protein (p-tau)[40], which synergistically impairs cognitive performance in patients with AD. Our findings showed that hippocampal and cortical p-tau levels in AD mice treated with VIP@-siBACE1 were lower than those in control AD mice treated with PBS (Fig. 5b and Supplementary Fig. S13b).

Vinpocetine has been reported to act synergistically with the BACE1 pathway by regulating glycogen synthase kinase-3β (GSK3β)[42]. Reasoning that the derivative of vinpocetine may elicit these pharmacologic functions, we undertook a validation study and discovered that p-GSK3β (Tyr216) led to an increase in stimulation of GSK3β in the AD group compared with that in the WT group, and VIP@siBACE1 and VIP@siScr treatment significantly decreased Tyr216 phosphorylation in the hippocampus and cortex of APP/PS1 mice compared with that in the AD group (Fig. 5c and Supplementary Fig. S13c). Therefore, VIP maintained the intrinsic inhibitory activity of GSK3β and BACE1 and exerted a synergistic anti-dementia therapeutic effect, which was consistent with the improvement in behavioral tests observed in APP/PS1 mice treated with VIP@siScr. Furthermore, the quantification of BACE1 gene expression showed that the BACE1 mRNA expression in the cortex of APP/PS1 mice was significantly inhibited after VIP@siBACE1 treatment (Fig. 5f). Combined with the Pearson correlation analysis of BACE1 gene expression levels with the protein levels with the cognitive and histopathologic improvement (Supplementary Figs. S14), the results showed that VIP could have additive effects beyond inhibiting BACE1, but silencing BACE1 was the main effect, rather than modulation of other pathways linked to neurodegeneration.

Oxidative stress[43] has a pivotal role in the initiation and progression of AD, and elevated levels of oxidative markers have been reported in the brain[44], blood, and cerebrospinal fluid of patients with AD. Neuroinflammation reactions[43] also contribute significantly to the pathogenesis of AD through elevating proinflammatory cytokines and continued aggregation of Aβ and p-tau. In our study, the levels of SOD, MDA, and core proinflammatory cytokines in APP/PS1 mice decreased after treatment with VIP@siScr and VIP@siBACE1, and the oxidative stress and inflammation environment in the brain was relieved (Fig. 5g, h); these findings might partially explain the downregulating effect of VIP and BACE1 protein on Aβ and phosphorylated tau levels. Therefore we used Nissl staining to assess neuronal damage in the brains of AD mice, focusing on the CA1 and CA3 regions of the hippocampus (commonly associated with memory)[45]. Nissl staining revealed deeper nuclear staining and cell body shrinkage in the CA1 and CA3 regions of the hippocampus in APP/PS1 mice treated with PBS and MC3@siBACE1, in contrast with findings in WT mice and those treated with VIP@siRNA (Fig. 5i). The results suggest that VIP@siRNA has neuroprotective effects. In conclusion, our use of the term 'multidimensional synergistic brain-targeted drug delivery system' reflects the ability of VIP to enact an integrated neuromodulatory function, inhibiting the GSK3β/BACE1 signaling cascade in addition to its conventional antioxidant and anti-inflammatory effects.

Routine tests of blood biochemistry, tissue sections, and hematotoxicity showed no significant differences between the PBS and VIP@siRNA treatment groups within 2 weeks after injection, indicating that VIP has suitable biocompatibility (Supplementary Figs. S15 and S16

and Table S7). We further observed no significant elevation in markers of immunogenicity, including complement and cytokines, in the VIP group. As shown in Fig. 5j, no significant change was observed in C5b9, C3a, and monocyte chemoattractant protein (MCP-1) levels at 2 weeks after treatment with VIP. However, an apparent variation above baseline was observed in the MC3 group. This is consistent with reports[30] that LNPs consisting of commercially available ionizable lipids, such as DLin-MC3-DMA, can activate the immune system, resulting in CARPA, an acute immunologic response that can lead to anaphylactic-like shock. Collectively, the aforementioned experimental results underscore the favorable safety profile of VIP.

## VIP as a delivery platform for brain disease drug delivery

We also investigated whether VIP loaded with different drugs could be used to treat diverse models of brain disease. In the first such model, we injected GL261-Luc cells into the brains of female mice for preliminary tests of siVEGF treatment (Supplementary Fig. S17a). Live imaging showed a reduction in real-time variation in luminescence after treatment with VIP@siVEGF. The quantitative results showed that the MFI of brain tumors in the VIP group was reduced by 18.88% at 3 d and 32.71% at 6 d after administration compared with the MC3 group (Supplementary Fig. S17b, c), showing the potential of delivering therapeutic siRNA drugs to brain tumors.

We further explored the delivery of small-molecule drugs and mRNA to a third model of fungal meningitis, thereby extending our validated animal models beyond mice bearing tumors and mice with dementia. The incidence of fungal meningitis in humans is increasing and is associated with high mortality; however, the clinical use of amphotericin B (AMB)[46], an antibiotic commonly used for epiphytes, is limited by its ineffective passage through the BBB. Here, we tested AMB-loaded LNPs (NP, GSH, VIP) with similar encapsulation efficiency (Supplementary Table S8) for the cumulative release of AMB (Supplementary Fig. S18) against cryptococcal meningitis in a female mouse model (Fig. 6a and Supplementary Fig. S19). We confirmed similar drug susceptibility of the *C. neoformans* and *C. neoformans*-Luc used in the in vivo tests with minimum inhibitory concentration assays (Supplementary Table S9). Live imaging showed a reduction in real-time variation in luminescence after administration of VIP@AMB. Moreover, quantitative results demonstrated that the MFI of the brain in the VIP group was reduced by 83.01% at 3 d and by 97.24% at 6 d after treatment, as compared with MFI in the commercially available control AmBisome group (Fig. 6b, c).

Increase in cerebral blood flow, reduction in brain fungal burden, and findings on histopathologic examination revealed that the extent of recovery in the VIP@AmB mice was similar to that in the positive control GSH@AMB mice and was better than mice treated with flucytosine, a BBB-penetrating antifungal agent typically used for cryptococcal meningitis; in contrast, the brain tissues of mice in the untreated, AmBisome, and VIP groups were heavily infiltrated with cryptococcus (Fig. 6d, e, Supplementary Fig. S20a, b and Supplementary Fig. S21). Moreover, the long-term survival of mice treated with VIP@AMB was significantly better than that of the other treatment groups (Fig. 6f and Supplementary Fig. S20c). Finally, intravenous injection of VIP@eGFP mRNA led to effective delivery of mRNA into the brain, and the fluorescence signal in the cerebrum and deep penetration of the brain parenchyma was better than those after injection of MC3@eGFP mRNA (Fig. 6g–i). Collectively, these results show that improving cerebral blood flow enhanced the probability of brain distribution and target-cell uptake of LNPs, improving targeted drug delivery and the corresponding therapeutic effects.

## Discussion

The natural product reservoir is a prominent source of abundant compounds that have had pivotal roles in drug discovery endeavors[47]. Of particular interest are small-molecule natural products,

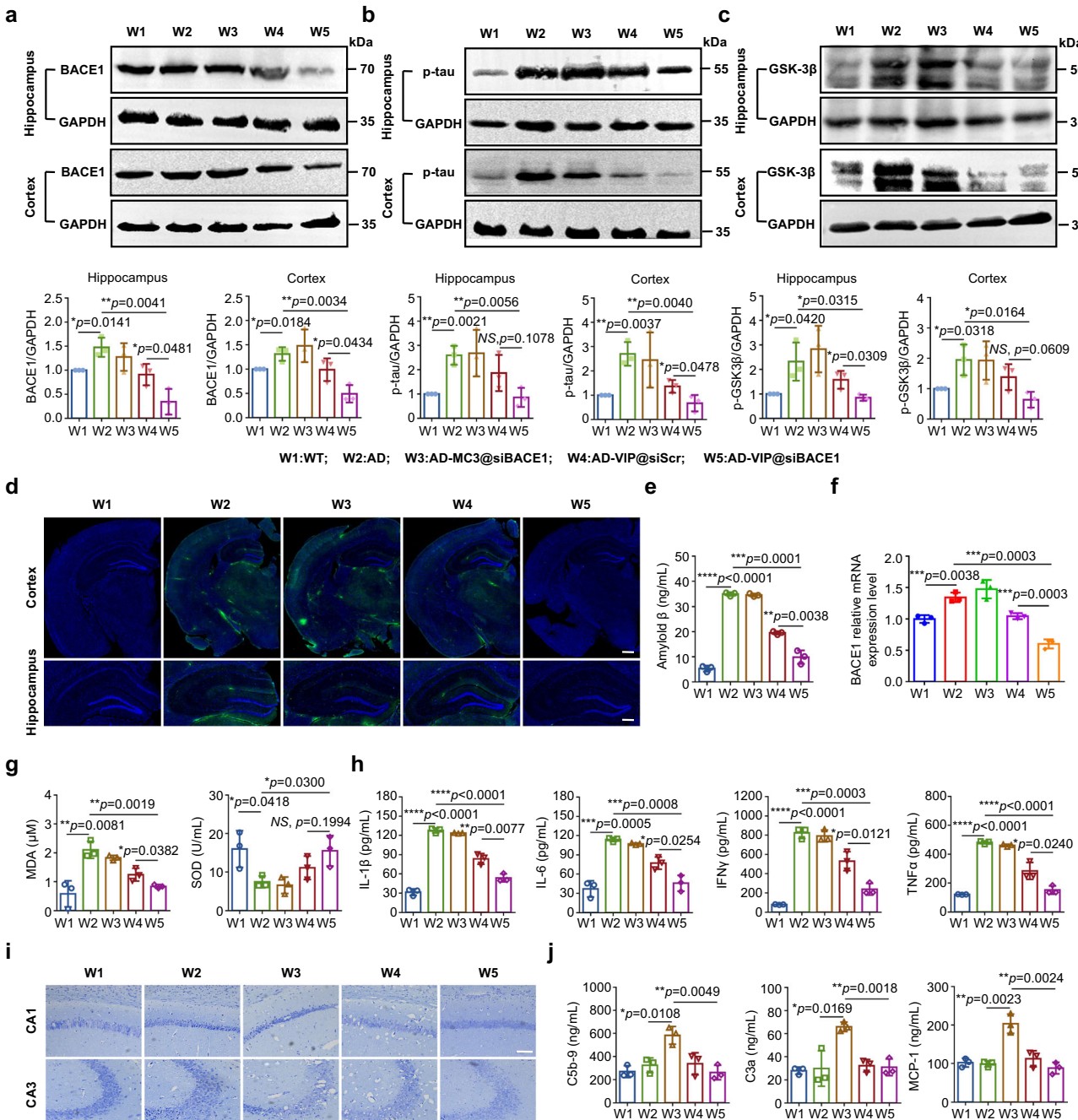

**Fig. 5 | Therapeutic synergy of VIP and siBACE1 to modulate AD hallmarks in APP/PS1 mice. a–c** Representative western blot data for BACE1, p-tau, and p-GSK3β protein expression in hippocampus and cortex from VIP@siBACE1-treated APP/PS1 mice, control APP/PS1 groups, and WT mice. Quantification of western blotting analysis of BACE1, p-tau, and p-GSK3β protein expression is shown relative to GAPDH. Data are presented as means ± SD (*n* = 3 biologically independent experiments). **d** Confocal laser scanning microscopy images of amyloid plaque burden; immunofluorescence of Aβ plaques (green) in hippocampus and cortex from APP/PS1 transgenic and WT mice. Nuclei were stained with DAPI (blue). Scale bar in top row = 500 μm, scale bar in bottom row = 250 μm. **e** ELISA evaluation of amyloid plaques in serum. Data are presented as means ± SD (*n* = 3 biologically independent samples). **f** BACE1 relative mRNA expression level in cortex was quantified by qPCR. Data are presented as means ± SD (*n* = 3 biologically independent samples).

**g, h** Oxidative stress markers (such as SOD and MDA) and proinflammatory cytokines (such as IL-1β, IL-6, IFN-γ, and TNF-α) in serum were assessed. Data are presented as means ± SD (*n* = 3 biologically independent samples). **i** Nissl staining of representative brain sections at 14 d after treatment with different formulations; scale bar = 40 μm. **j** ELISA evaluation of complement activation-related pseudoallergy (CARPA) markers such as C5b9, C3a, and monocyte chemoattractant protein-1 (MCP-1) in serum. Data are presented as means ± SD (*n* = 3 biologically independent samples). All samples were collected after seven injections of LNP. *P < 0.05, **P < 0.01, ***P < 0.001, ****P < 0.0001, NS means no significance. Statistical significance was calculated with two-tailed unpaired *t*-tests. Data in (**d, i**) are representative of two independent experiments with similar results. Source data are provided as a Source Data file.

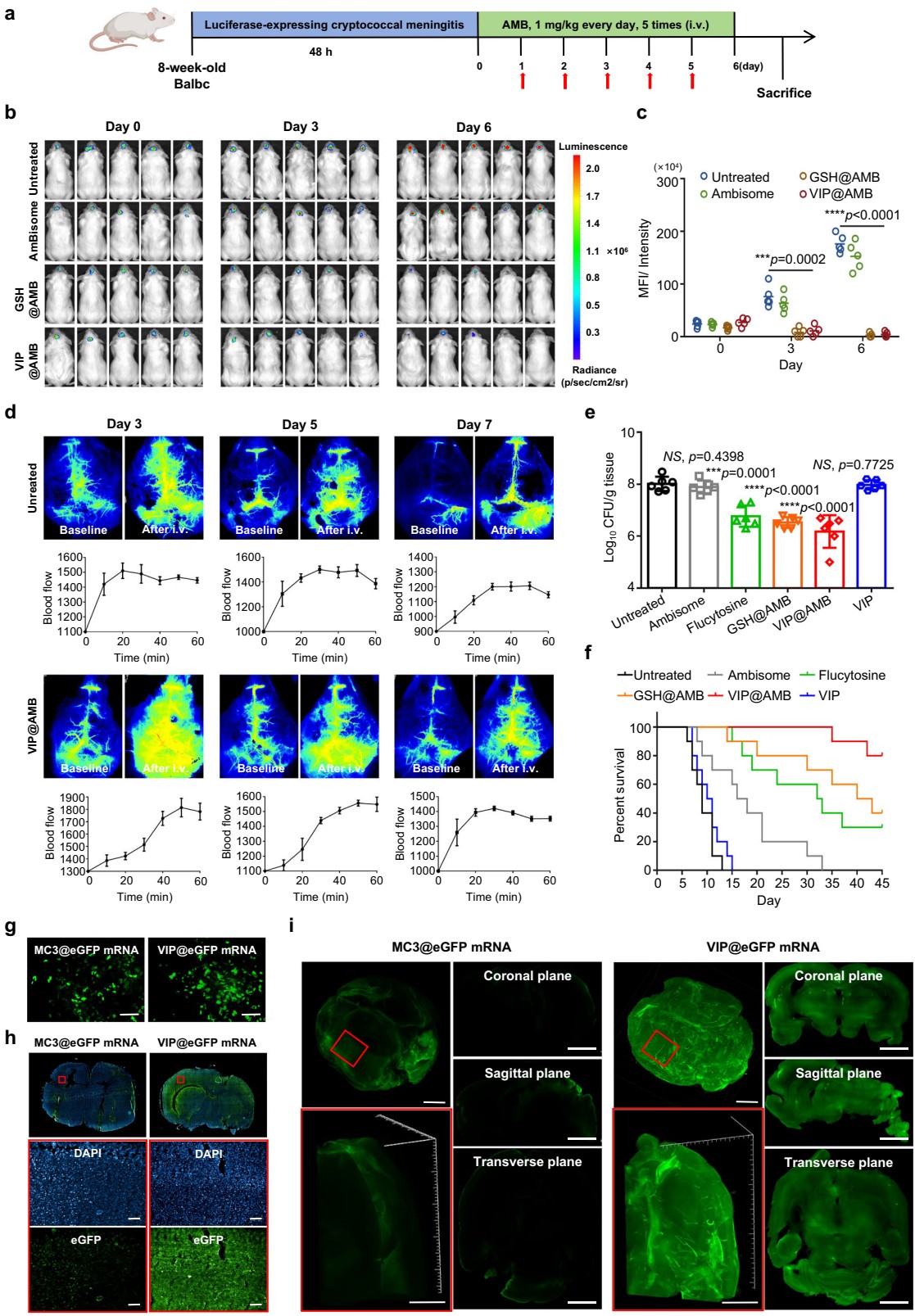

characterized by their extensive scaffold diversity and pharmacophore patterns, which make them promising lead compounds during the drug development process[48]. Recent advances have shed light on different perspectives in drug delivery, exemplified by the case of (−)-epigallocatechin-3-O-gallate (EGCG), a major constituent of green tea that has anticancer[49], antiviral[50], and neuroprotective[51] effects. Complexing monoclonal antibodies targeting HER2/neu (erbB2) with

EGCG was shown to confer protection to the proteins during drug delivery by forming a stable and reversible structure through hydrophobic interactions. This formulation has demonstrated superior cancer-inhibitory effects relative to free protein, owing to synergistic interactions[52]. Another example is tannic acid, a plant-derived flavonoid with a high affinity for elastins and collagens, the principal components of the myocardium. TANNylating therapeutic proteins or

**Fig. 6 | In vivo evaluation of the efficacy of VIP-NP for the delivery of small-molecule and macro-molecule drugs. a** Schematic of the experimental timeline for treating a mouse model of cryptococcal meningitis. AMB, amphotericin B. (Created with BioRender.com). **b, c** Qualitative and quantitative analysis of live imaging of *C. neoformans*-Luc infected mice. Data are presented as means ± SD (n = 5 biologically independent samples). **d** Blood flow enhancement effect on brain microvessels detected by using laser speckle flowmetry in brain *C. neoformans* infected model on the 3, 5, 7 day after infection treated with PBS or VIP@AMB as above. Data are presented as means ± SD, were obtained by measuring the blood perfusion unit every 10 min. **e** Fungal colony burdens in the brain tissue of *C. neoformans* infected mice in each group after treatment. Data are presented as means ± SD (n = 6 biologically independent samples). **f** Survival curves of each treatment group. (n = 10 biologically independent animals). **P < 0.01, ***P < 0.001, ****P < 0.0001, NS means no significance. Statistical significance was calculated with **e** two-tailed unpaired t-tests and **c** multiple t-tests. **g, i** VIP effectively delivers eGFP mRNA in vitro and in vivo. **g** Fluorescence microscope images of eGFP mRNA transfected in vitro, scale bar = 50 μm. **h** Fluorescence imaging of brain tissue sections after administration of 10 μg eGFP mRNA, scale bar = 100 μm. **i** 3D hyalinization microscope images of brain tissue processed by PEGASOS tissue-clearing protocols after administration of 10 μg eGFP mRNA, scale bar = 1500 μm. Data are representative of three (**g, h**) and two (**d, i**) independent experiments with similar results. Source data are provided as a Source Data file.

peptides can enhance the ability to specifically target heart tissue via systemic injection[53]. Ginsenosides, naturally occurring amphiphilic molecules structurally resembling the hydrophobic triterpene skeleton of cholesterol, and bearing hydrophilic glycosyl moieties, have been used to construct a liposome having tumor-targeting and immune-regulating capabilities. Substituting cholesterol with the ginsenoside Rg in the lipid bilayer led to a multifunctional liposome system[54]. Incorporating pharmacologically active natural small molecules in drug delivery systems can circumvent the complexities associated with formulation design, enabling the development of versatile delivery systems capable of meeting diverse requirements in treating complex diseases[55]. Notably, most of these natural products have undergone extensive study and validation as drugs[56] or food additives[57], endowing them with heightened clinical translational value owing to their established safety profiles and "druggability", in contrast to unexplored chemical structures. Alkaloids, nitrogen-containing organic compounds with substantial biological activity[56], have been the subject of extensive research that led to the discovery of numerous alkaloidal drugs since their approval by the US Food and Drug Administration, including the antimalarial agent quinine and the analgesic morphine. In this study, we used vinpocetine alkaloid, a specific regulator of cerebral blood flow, to meticulously design an efficient brain-targeted drug delivery system to address challenges associated with distribution and drug accumulation for cerebrovascular lesions. This innovative endeavor to leverage alkaloids in drug delivery, in particular the delivery of nucleic acids, further underscores the significance of ongoing exploration of natural products in constructing efficient drug delivery systems tailored to diverse demands.

Currently, most ionizable lipid designs incorporate tertiary amine groups to achieve pH sensitivity and facilitate intracellular endosome escape[58]. However, even though the cyclic tertiary amine head group exhibits specific steric hindrance, it may exhibit superior binding interactions with nucleic acids compared with the simpler linear tertiary amine head group. For example, C12-200, which contains piperazine, has demonstrated the ability to simultaneously deliver five types of hepato-targeted siRNAs into cells[59], whereas A18-Iso5-2DC18@mLuc, which also contains piperidine, leads to twofold higher protein expression levels compared with commercially available LNPs formulated with Dlin-KC3-DMA[60]. Nevertheless, synthetic lipidoids do not always have desirable biological activities, despite their ability to facilitate RNA encapsulation and cellular delivery. Cellular toxicity and immunogenicity are significant drawbacks associated with the use of cationic LNPs, particularly when repeated administration is required[61]. Alkaloids, a class of nitrogen compounds with heterocyclic structures and significant biological activity, warrant attention[62]. Vinpocetine, in particular, displays anti-inflammatory and antioxidant effects, offering the potential for reducing the toxicity of certain chemotherapy drugs, such as cisplatin-induced acute kidney injury[63] and etoposide-induced brain injury[64]. Moreover, vinpocetine has shown promise in mitigating SARS-CoV-2-induced hyper-inflammatory and oxidative stress[65]. Experimental results indicate that combining VIP, which are vinpocetine-derived ionizable-lipidoid nanoparticles, with RNA drugs represents a safe, non-

liver-targeted strategy that enables sufficient knockdown of disease-causing genes with lower intrinsic toxicity and immunogenicity. Further, the inherent pharmacologic effects of VIP contribute to the intended therapeutic response and mitigate the acute immune response associated with RNA therapeutics. The introduction of VIP lipidoids represents the application of the multiple fused polycyclic ring structure based on alkaloids in the design and implementation of ionizable lipid head groups. This innovation has led to the development of a straightforward, versatile, and effective multifunctional carrier platform with extensive potential applications in drug delivery.

In summary, we successfully developed a VIP (vinpocetine-derived ionizable-lipidoid nanoparticles) delivery system, which was remarkably effective at facilitating brain-targeted drug delivery that was safe and showed pharmacologic synergy. This study offers valuable insights into the development of a highly promising 'active distribution' strategy through selective modulation of organ blood flow. Moreover, our findings shed light on the untapped potential of alkaloid natural products in the field of drug delivery and precision medicine.

## Methods

### Mouse studies
All animal studies were approved by the IACUC (Institutional Animal Care and Use Committee) of Southwest University Laboratory Animal Center (IACUC Issue No. IACUC-20221114-14). All animal experiments were conducted under the guidelines of the Ethical Review Committee of Experimental Animals at the Southwest University of China. In consideration of achieving gender balance across various animal models to ensure the universality of targeting and efficacy, we adopted male APP/PS1 mice for the AD model and female mice for the brain tumor and meningitis models to comprehensively evaluate the VIP system.

### Preparation of VIP
VIP were produced by using microfluidic preparation of vinpocetine derivatives from the molecular library. Lipids were dissolved in ethanol at varying molar ratios (ionizable lipids: helper lipids: cholesterol: PEG-DMG$_{2000}$). The ionizable lipids consisted of all vinpocetine derivatives in the molecular library, whereas the helper lipids included DOPE, DOPC, and DSPC. Specific combinations of these lipids are listed in Supplementary Table S1. The lipid mixture was combined with a 6.25 mM sodium acetate buffer (pH=4) containing siRNA at a ratio of 3:1 (water: ethanol) by using a microfluidic mixer (AITESEN, China). The preparations were then dialyzed overnight in PBS (pH 7.4) with dialysis bags. Finally, the preparations were concentrated with an ultra-centrifugal filter (Millipore, Billerica, MA, USA).

### Characterization of physiochemical properties of VIP
The particle size and zeta potential of VIP, both before and after encapsulation with siRNA, were measured at room temperature by dynamic light scattering (DLS) (Zetasizer, Malvern, England). The structure of LNPs was observed by transmission electron microscopy (HITACHI, Japan), and the concentration of siRNA encapsulated in

LNPs was determined by using the Quant-iT RiboGreen RNA Assay kit. The procedures were carried out according to standard protocols. For the detection of free siRNA, the LNP was diluted in Tris-ethylenediaminetetraacetic acid (EDTA) buffer and dispensed into a 96-well plate. For the detection of total siRNA, LNPs were also diluted in Tris-EDTA buffer containing 2% Triton X-100 and incubated for 30 min. RiboGreen working solution was added to the corresponding wells, and the fluorescence intensity was measured by using a microplate reader to determine the amount of free siRNA and siRNA after dialysis. The concentration of siRNA was calculated from standard curves, and encapsulation efficiency was calculated with the following formula: encapsulation efficiency (%) = [m(total siRNA - free siRNA)/m(total siRNA) × 100].

### Determination of the acid dissociation constant (pKa)

VIP were prepared and diluted to approximately 2 mM in PBS and then further diluted to 100 µM in a buffer containing 130 mM NaCl, 100 mM ammonium acetate, 10 mM HEPES, and 10 mM morpholineethanesulfonic acid (MES). The pH of each portion was adjusted to between 3 and 10 by using NaOH or HCl. 2-(*p*-toluidine) -6-naphthalene sulfonic acid (TNS) was dissolved in distilled water to a concentration of 100 µM and added to pH-adjusted VIP solutions to achieve a final concentration of 1 µM. The fluorescence intensity of each formulation was determined at room temperature by using a Multimode Microplate Reader (BioTek Synergy H1, USA) with excitation wavelength of 321 nm and emission wavelength of 445 nm. The pKa was identified as the pH at which the fluorescence intensity reached half its maximum value.

### Measurement of cerebral blood flow

After being anesthetized with isoflurane, mice were secured in a stereotactic frame (RWD Life Science Co., Ltd, China). An incision was made along the midline of the skull to expose the skin, and laser speckle contrast imaging (RWD Life Science Co., Ltd, China) was used to measure cerebral blood flow. Laser speckle blood flow images were recorded, regions of interest were identified, and mean blood flow index within these regions was calculated in real-time.

### Surface plasmon resonance binding assays

The binding kinetics of PDE1 and BSA protein were determined by using surface plasmon resonance binding (SPR) assays (Nicoya Lifescience, Waterloo, Canada). The running buffer for these experiments consisted of 40 mM 4-Morpholinepropanesulfonic acid (MOPS), 15 mM MgCl$_2$, 150 mM NaCl, 0.5 mM EDTA, 0.05% Tween 20, and 5% dimethylsulfoxide. Before the SPR measurements, the COOH sensor chip was installed, and the system was filled with running buffer at a constant flow rate of 150 µL/min. To remove bubbles from the system and stabilize the drift signal with running buffer, 80% iso-propyl alcohol was imputed into the entire pipeline. PDE1 or BSA was immobilized on the COOH sensor chip at a concentration of 25 µg/mL by using a standard amine coupling procedure. After surface activation with 0.2 M 1-(3-Dimethylaminopropyl)-3-ethylcarbodiimide hydrochloride (EDC) and 0.05 M N-hydroxysuccinimide (NHS), the protein was injected at a constant flow rate of 20 µL/min for 5 min to achieve an immobilization level of 2000 response units, with the remaining activated groups being blocked by ethanolamine. The analyte was injected at a flow rate of 20 µL/min, and dissociation was monitored for 7 min before the chip was regenerated with 10 mM HCl. The chip was then equilibrated with running buffer for another 7 min before the next injection. Affinity constants (K$_D$) were calculated with Tracedrawer evaluation software by fitting the data to a kinetic model.

### Gel retardation assay

siRNA (1 OD) was dissolved in 125 µL of diethyl pyrocarbonate-treated water and mixed with VIP solutions at siRNA/LNP weight ratios of 5:1, 1:1, 1:2, 10:1, 1:5, 1:10, 1:15 and 1:20. The mixture was incubated for 30 min at room temperature. The siRNA binding ability of VIP was analyzed with agarose gel. Lipid formulations containing 500 ng of siRNA were combined with 6× loading buffer and loaded onto a 2% agarose gel containing 0.02% gel stain. Electrophoresis was performed at a voltage of 180 V for 15 min in 1× tris-acetate-EDTA running buffer, and the results were recorded by using a gel image analysis system (Tanon, China).

### Intracellular transfection behavior and mechanism

bEnd.3 cells were seeded in plates and incubated in medium at 37 °C in 5% CO$_2$ for 12 h. For cellular transfection, the cells were treated with different LNP@FAM-siRNA formulations by replacing the medium with medium lacking FBS and incubating for an additional 4 h. At the end of the assay, cells for quantification were digested with trypsin (0.25%), washed three times with PBS, resuspended in 300 µL of PBS, and analyzed with a flow cytometer (FACSverse, BD) and FlowJo 7.6 software. For intracellular behavior, the cells were stained with Lysotracker Red (KeyGEN BioTECH, China) at 37 °C for 2 h and then treated with different LNP@FAM-siRNA formulations by replacing the medium with medium lacking FBS and incubating for an additional 4 h. At the end of the assay, the cells were washed three times with PBS and stained with Hoechst 33342 (Beyotime, China) at room temperature for 10 min. Images were then acquired with a laser confocal microscope with an oil-immersion objective (Zeiss LSM780, Germany), and colocalization analysis was done with Zeiss ZEN software. For the mechanism of transfection, the cells were treated with chlorpromazine (20 µM), filipin (1.25 µM), monensin (35 µM), brefeldin A (100 µM), amiloride (200 µM), M-β-CD (20 mM) or fresh DMEM for 30 min. LNP@FAM-siRNA was added to the corresponding wells, and the cells were incubated at 37 °C for an additional 4 h. At the end of the assay, cells were digested with trypsin (0.25%), washed three times with PBS, resuspended in 300 µL of PBS, and collected with a flow cytometer (FACSverse, BD) and analyzed with FlowJo 7.6 software.

### Analysis of reactive oxygen species levels in single cells

To assess oxidative stress and intracellular reactive oxygen species (ROS) levels in cells, a reactive oxygen species assay kit was used to detect ROS as green fluorescent signals of DCFH-diacetate (DA) (Beyotime, China) as follows. bEnd.3 cells from each treatment group were incubated at 37 °C for 30 min in medium lacking FBS that contained 10 µM DCHF-DA and then replaced with fresh medium with ionizable molecules at doses of 0.25 mg/mL (and incubated for 2 h) or 0.05 mg/mL (and incubated for 1 h, 4 h, 12 h or 24 h). The cells were then washed three times in PBS containing 0.1% BSA before being mounted on glass slides with PBS for microscopy. Fluorescence intensity in individual cells was measured with a real-time single-cell multimode analyzer equipped with optical fiber probes (Rayme, China). Photon counts were continuously detected and calculated during a 150-s period after the tips were inserted into an immobilized cell in a PBS droplet[66].

### In vitro cytotoxicity assay

bEnd.3 cells were seeded at a density of 5000 cells per well in 96-well plates and cultured in DMEM with 10% fetal bovine serum and 1% penicillin/streptomycin at 37 °C in 5% CO$_2$ for 12 h. Subsequently, different LNPs were added to the cells and incubated for an additional 48 h. At the end of the assay, MTT solution (5 mg/ml) was added, and the samples were incubated at 37 °C for 4 h. Cell viability was determined by measuring the absorbance of the extracellular medium at 570 nm.

### Cellular uptake of liposomal formulations

Cells (bEnd.3, A172, and PDE1$^{-/-}$ A172) were seeded in plates and incubated in medium at 37 °C in 5% CO$_2$ for 12 h. Thereafter, the cells were treated with different LNP@DiD formulations by replacing the medium with fresh medium and incubating for an additional 4 h. At the

end of the assay, cells for quantification were digested with trypsin (0.25%), washed three times with PBS, resuspended in 300 μL of PBS, and collected with a flow cytometer (FACSverse, BD) and analyzed with FlowJo 7.6 software.

## Measurement of the cellular accumulation of rhodamine 123

The accumulation of rhodamine 123, a fluorescent substrate of P-glycoprotein, in BMECs was measured by incubating BMECs with 20 μM rhodamine 123 in the absence or presence of VIP for 1 h at 37 °C. At the end of the assay, cells were digested with trypsin (0.25%), washed three times with PBS, resuspended in 300 μL of PBS, and collected with a flow cytometer (FACSverse, BD) and analyzed with FlowJo 7.6 software.

## In vitro blood-brain barrier penetration assay

An in vitro BBB model was established by seeding brain microvessel endothelial cells (BMECs; 10,000 cells per well) on the upper chambers of 24-well Transwell plates (Corning, USA) and culturing them to form monolayers. The culture medium was replaced every 2 days, and the integrity of this bilayer was monitored by measuring transendothelial electrical resistance (Millicell-ERS, Millipore, USA). When the transendothelial electrical resistance value reached more than 150 Ω·cm2, the medium was replaced, and different LNPs labeled with DiD were added to the upper chambers of the Transwells and incubated for 0.5 h, 1 h, 2 h, 4 h, or 8 h. The Transwell membranes were transferred onto glass microscope slide;s, and the nuclei of the cells were stained with Hoechst 33342. The penetration of LNPs was observed by a high-content analysis system (Operetta CLS, PerkinElmer, USA) and reconstructed three-dimensional images by Imaris software. The medium in the basolateral chamber was collected, and the fluorescence was analyzed with a Multimode Microplate Reader (Agilent BioTek Synergy H1, USA) with an excitation wavelength of 485 nm and an emission wavelength of 535 nm.

## In vivo imaging

For the brain-targeting assay, LNP@DiD was injected into healthy mice via the tail vein. Fluorescence images were acquired at predetermined time points (0.5 h, 1 h, 2 h, 4 h, and 8 h) by using a VISQUE in vivo Smart-LF System (Vieworks, Korea). Mice were euthanized at predetermined time points, and their major organs were separated for ex vivo imaging.

A brain tumor model was established by injecting a cell suspension into the brains of mice as follows. Mice were anesthetized and fixed in a stereotactic device before injection of 5 μL of GL261-Luc (50,000 cells) into the right brain over a period of 3 min and then slowly withdrawn. At 7 d after the operation, mice were intravenously injected with LNP@siVEGF (1 mg/kg) every day for 5 days, and images obtained with a Lumina III Imaging System (PerkinElmer, USA) at 0 d, 3 d, and 6 d after treatment.

A meningitis model was established by injecting a fungal suspension into the brains of mice as follows. Mice were anesthetized and fixed in a stereotactic device before injection of 5 μL of luciferase-expressing strain of *C. neoformans* (500,000 colony-forming units [CFU]/mL) into the right brain over a period of 3 min and then slowly withdrawn. At 48 h after the operation, mice were intravenously injected with LNP@AmB at a dose of 1 mg/kg amphotericin B every day for 5 days and images were obtained with a Lumina III Imaging System (PerkinElmer, USA) at 0 d, 3 d, and 6 d after treatment.

## Novel object recognition test

The NOR test was performed according to published methods[38]. The experimental apparatus consisted of a white rectangular open field box made of polyethylene (50 cm by 50 cm by 50 cm). Mice were habituated to the experimental apparatus by being exposed to it for 10 min in the absence of objects on the day before training. During the training phase, mice were placed in the experimental apparatus with two identical objects (odorless cuboids were used to prevent mice

from climbing onto the objects and to avoid the preference for sitting on them) and allowed to explore for 10 min. After 24 h, mice were placed back in the apparatus where one of the objects had been replaced with a novel object and allowed to explore for another 10 min. The discrimination index (DI) and preference index (PI) were used to assess NOR and account for differences in exploration time. DI and PI were calculated based on the time spent exploring the objects (sniffing, trying to move, and front paw pushing were considered exploring, but time spent near the objects without investigation or passing by them was not). Data were collected by using tracking software EthO Vision XT8.5, and behaviors were manually scored from videos. DI was calculated as the time spent exploring the novel object minus the time spent exploring the familiar object divided by the total exploration time. PI was calculated as the proportion of total time spent exploring either the new or old object. $[DI = T_{novel} - T_{familiar}/(T_{novel} + T_{familiar})$, $PI = T_{novel}$ or $T_{familiar}/(T_{novel} + T_{familiar})]$. All DI values fall between $-1$ and $+1$, whereas PI values fall between 0 and 1.

## Morris water maze test

We used the MWM test to evaluate spatial learning and memory according to standard protocols[39]. A pool was divided into four quadrants, with a different symbol (pentagram, square, triangle, and circle) affixed to the wall of each quadrant to provide extra maze spatial cues. The water temperature was maintained at $22 \pm 1$ °C, and highly dispersed food-grade titanium dioxide was added to aid mouse tracking. All MWM experiments were conducted daily at the same time in the afternoon in a confined space without noise or strong light sources. Mice were habituated to the room for 2 h before the experiment. Each mouse was trained to find a hidden platform for 5 consecutive days, with 4 trials per day and a 20- to 30-min intertrial interval. Mice were placed in the water facing the wall of the pool, and the platform was randomly allocated to one of the four quadrants. The time taken for the animals to find the platform was recorded. If the latency to find the platform exceeded 60 s during training sessions, animals were guided to the platform and kept there for 10 s. Mice were trained for 5 days to find the platform. At 24 h after training, the platform was removed, and the 60-s probe test was conducted as follows. Mice were placed in the water facing the quadrant opposite the target quadrant, and the time spent in the target quadrant and the number of crossings of the platform location were recorded as indicators of spatial memory. Data were collected by using tracking software EthO Vision XT8.5.

## Western blotting

For therapeutic evaluations, mouse brain tissues were collected after the completion of behavioral assessments as follows. Animals were euthanized and transcardially perfused with saline before tissue samples (whole hippocampus and cortex) were removed and homogenized in lysis buffer with 1% phosphatase inhibitors and 1% phenylmethylsulfonyl fluoride (Beyotime, China) and centrifuged for 15 min ($13,000 \times g$, 4 °C). The protein concentration of the supernatant was determined with a BCA Protein Assay Kit (Beyotime, China). Approximately 20 μg of protein was loaded and separated by 10% SDS-polyacrylamide gel electrophoresis before being transferred to a polyvinylidene fluoride membrane for blotting. The transferred membrane was blocked with fresh blocking buffer containing 5% nonfat dry milk in Tris-buffered saline at 37 °C for 1 h. Different primary antibodies were added to the membrane and incubated overnight at 4 °C before being incubated with appropriate secondary antibodies for 1 h at 37 °C. Blots were developed by using the ECL technique (Beyotime, China), with GAPDH as an internal control for total protein. Primary antibodies against BACE1 (Abcam2), p-tau (Thermo Fisher Scientific), p-GSK3β (Thermo Fisher Scientific), or GAPDH (Proteintech) and HRP-conjugated IgG rabbit secondary antibodies were used. Data were quantified with ImageJ software, and the results were recorded on a gel image analysis system (Tanon, China). Unprocessed

scans of the most important blots were provided in the Source Data file.

## Real-time PCR

Endogenous BACE1 gene silencing activity of LNP@siBACE1 was investigated by quantitative real-time PCR (qRT-PCR). After their 2-week treatments, mice were anesthetized and transcardially perfused with saline. Total RNA extraction, reverse transcription, and qPCR were carried out by kit protocols (Accurate Biotechnology) and detected with the ROCGENE Archimed RT-PCR System. The primer sequences were as follows: BACE1 forward primer: 5′-TACTA CTGCCCGTGTCCACC -3′ and reverse primer: 5′- ACAACCTGAGGGG AAAGTCC -3′. GAPDH was used as reference housekeeping gene with forward primer 5′- TTGATGGCAACAATCTCCA-3′ and reverse primer: 5′- CGTCCCGTAGACAAAATGGT-3′. The relative mRNA expression level was calculated based on the comparative $Ct$ method.

## Enzyme-linked immunosorbent assay

To evaluate the potential toxicity of LNP to the immune system, serum samples were obtained from APP/PS1 mice after injection of each LNP@siBACE1 formulation once every 2 days for 2 weeks. Samples were collected from the retro-orbital plexus, and plasma was separated by centrifugation at $2200 \times g$ for 10 min at 4 °C. Complement activation-related pseudoallergy (CARPA) parameters in the supernatant, including complement C5b9, complement C3a, and MCP-1 levels, were quantified with ELISA kits.

To evaluate the synergistic effect of LNP@siBACE1 formulations on AD models in vivo, each formulation was injected into APP/PS1 mice every 2 days for 2 weeks and serum was collected for evaluation. Blood samples were collected from the retro-orbital plexus, and plasma was separated by centrifugation at $2200 \times g$ for 10 min at 4 °C. Levels of β-amyloid (Aβ) and inflammatory cytokines (e.g., IL-1β, IL-6, IFNγ, and TNFα) were measured with ELISA kits.

## Blood biochemistry examinations

To evaluate the potential toxicity of LNP to the liver and kidney, mice were subjected to serum biochemical evaluations as follows. The mice were injected with each LNP@siBACE1 formulation once every 2 days for 2 weeks. Blood samples were collected from the retro-orbital plexus, and plasma was separated by centrifugation at $2200 \times g$ for 10 min at 4 °C. The plasma was then analyzed for biochemical parameters, including alkaline phosphatase (ALP), alanine aminotransferase (ALT), aspartate aminotransferase (AST), uric acid (UA), urea (UREA), and creatinine (CREA).

## Histologic staining

After treatment, APP/PS1 mice were euthanized and perfused with saline. Brains were collected and fixed in 4% paraformaldehyde for 24 h, dehydrated, frozen with OCT, and cut into 20 µm frozen slices by using a freezing microtome (Leica, Germany). For immunofluorescence, the sections were fixed with 4% paraformaldehyde solution for 30 min and washed three times with PBS. The sections were permeabilized with 0.1% Triton X-100 in PBS for 15 min, blocked with 5% BSA for 2 h, and incubated with anti-β-amyloid primary antibody (Bioss) overnight at 4 °C. Sections were then washed three times with PBS and incubated with FITC-conjugated goat anti-rabbit IgG (Bioss) for 2 h at room temperature, after which the slices were stained with DAPI for 10 min. Fluorescence images were obtained with Olympus (Japan). For Nissl staining, the brain sections were stained with cresyl violet to observe neuronal injury. The hearts, lungs, livers, spleens, and kidneys were also collected and fixed overnight in 4% paraformaldehyde. Serial sections were collected and stained with hematoxylin and eosin to examine histologic changes in the organs. Images were acquired with an inverted fluorescence microscope (DMi8, Leica, Germany).

The brain-infected mice were also euthanized and perfused transcardially. The brains were collected and fixed overnight in 4% paraformaldehyde. Serial coronal brain sections (20 µm) were collected and stained with Groctt-Gomori methenamine silver to examine histologic changes in the brain. Images were acquired with an inverted fluorescence microscope (DMi8, Leica, Germany).

## In vitro release of amphotericin B

In vitro release of amphotericin B (AMB) from AMB-loaded LNPs (NP, GSH, VIP) was evaluated by the dialysis bag method. Briefly, 1 mL LNP sample was transferred into a dialysis bag with 8000-14000 Da, and immersed in 40 mL of PBS (containing 0.1% sodium deoxycholate) under continuous stirring at 37 °C. The entire volume of the release medium was measured at predetermined time points (0.5 h, 1 h, 2 h, 4 h, 8 h, 12 h, 24 h, 48 h, and 72 h) and replaced with a fresh and equivalent volume medium. The samples were centrifuged at $11000 \times g$ for 10 min and then analyzed by HPLC.

## Minimum inhibitory concentration assay

*C. neoformans* and *C neoformans* labeled with a luciferase tag, -Luc, were cultivated and diluted to concentration at $1 \times 10^4$ CFU/mL with YPD and 1640 medium, respectively. The fungal suspensions were then transferred into a 96-well plate at 100 µL/well, and 100 µL drug solution at a concentration ranging from 0-1024 µg/mL was added to corresponding wells. The plates were incubated for 48 h at 30 °C and the OD value was measured at 490 nm by a Multimode Microplate Reader (BioTek Synergy H1, USA, Switzerland). The MIC was defined as the lowest concentration of test agents that inhibited more than 90% growth compared with the control group.

## Tissue burden study

Mice were anesthetized and fixed in a stereotactic device before the injection of 5 µL of *C. neoformans*(-Luc) (500,000 CFU/mL) into the right brain over a period of 3 min and then slowly withdrawn. At 48 h after the operation, mice were intravenously injected with LNP@AMB at a dose of 1 mg/kg amphotericin B or 25 mg/kg flucytosine every day for 5 days. On day 6, all mice were euthanized, and brain tissues from each group were carefully removed, weighed, and homogenized with sterile saline in a ratio of 1 g organ to 3 mL of saline. The homogenate was serially diluted with sterile saline, and then 30 µL of the suspension was spread on yeast extract–peptone–dextrose medium and incubated at 30 °C for 48 h to determine the number of CFU per g of brain tissue.

## Survival rate study

The brain-infected mouse model was induced as described above. At 48 h after inoculation, the infected mice were randomly divided into four groups and treated intravenously with saline and different LNP@AMB at a dose of 1 mg/kg amphotericin B or 25 mg/kg flucytosine every day for 5 days. Survival rates of the model mice were monitored for 45 days after treatment.

## mRNA transfection

Cells were seeded in a 12-well dish at a confluence of 60%–70% and transfected with eGFP mRNA formulated under different LNPs in DMEM without FBS for 4 h at 37 °C. Cells were cultured for another 20 h after FBS supplementation and viewed with a fluorescence microscope (DMi8, Leica, Germany).

## Fluorescence imaging

For in vivo mRNA transfection, mice were intravenous injected with LNP@eGFP mRNA (1 mg/kg), and 24 h later all mice were euthanized, and their brains were separated and fixed in 4% paraformaldehyde for 24 h, dehydrated, frozen with OCT compound, and cut into 20 µm frozen slices by a freezing microtome (Leica, Germany). Sections were

fixed with 4% paraformaldehyde solution for 30 min and washed three times with PBS, stained with Hoechst 33342 for 10 min, and fluorescence images were obtained with an Operetta CLS high-content analysis system (PerkinElmer, USA).

## PEGASOS tissue-clearing protocols and imaging

The clearing solutions were prepared according to standard protocols[67] as follows. For decolorization solutions, Quadrol was diluted with water to a final concentration of 25% v/v and ammonium was diluted with water to a final concentration of 5% v/v. For gradient tB delipidation solutions, pure tert-butanol (tB) was diluted with distilled water to prepare gradient delipidation solutions: 30% v/v, 50% v/v, and 70% v/v. Quadrol was then added at a final concentration of 3% w/v to adjust the pH to above 9.5. For tB-PEG dehydration solutions, the dehydrating solution was composed of 70% v/v tert-butanol, 27% v/v PEG methacrylate $M_n500$ (PEGMMA$_{500}$), and 3% w/v Quadrol. For the BB-PEG clearing medium (refractive index R.I. 1.543), BB-PEG was prepared by mixing 75% v/v benzyl benzoate and 25% v/v PEGMMA$_{500}$ supplemented with 3% w/v Quadrol together.

To clear brain tissue samples, samples were fixed with 4% paraformaldehyde at 4 °C for 24 h and then treated with Quadrol decolorization solution at 37 °C for 2 days. Samples were then immersed in gradient delipidation solutions at 37 °C in a shaker for 2 days, followed by dehydration solution treatment for 2 days and BB-PEG clearing medium treatment for 5–7 days until reaching transparency. Samples were then preserved in the clearing medium at room temperature.

The cleared brains were imaged with a light-sheet microscope (LiToneXL, Light Innovation Technology, China) equipped with a 43× objective lens (numerical aperture = 0.28, working distance = 20 mm). Thin light sheets were illuminated from the four sides of the samples, and a merged image was saved.

## Statistical analysis

Data are presented as means ± SD and statistical significance was determined by GraphPad Prism software 8. The differences between the groups were determined by using two-tailed unpaired $t$-tests and multiple $t$-tests. The threshold for statistical significance was $P < 0.05$ with 95% confidence intervals. In all figures, where $p < 0.0001$ is indicated, the $p$-value was too low for Prism software to provide an exact value. Association between BACE1 and p-tau, p-GSK3β, and Aβ protein expression in brain tissues was calculated using Pearson correlation test.

## Reporting summary

Further information on research design is available in the Nature Portfolio Reporting Summary linked to this article.

## Data availability

The data supporting the findings of this study are available within the paper and its supplementary information. Source data are provided with this paper. Any additional raw data will be available from the corresponding authors upon reasonable request. Source data are provided with this paper.

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

## Acknowledgements

The authors would like to thank Christine Wogan, MS, ELS, of MD Anderson Cancer Center's Division of Radiation Oncology for editorial assistance. This work was supported by the National Key Research and Development Program of China (2023YFF0724202), National Natural Science Foundation of China (NSFC No. 82373808, 82073789), Chongqing Science Fund for Distinguished Young Scholars (CSTB2023NSCQ-JQX0021), Fundamental Research Funds for the Central Universities (SWURC2020001), the project for Chongqing University Innovation Research Group, Chongqing Education Committee

(CXQT20006) and the State Key Laboratory of Natural and Biomimetic Drugs (K202207) to C.L.The authors wish to acknowledge Dr. Huan Zhao (Revvity) and Xiaogang Wang (Revvity) for their advice on experimental design and helpful discussions. The authors would also like to thank the Academy of Agricultural Sciences, Southwest University, for providing access to instrumentation and technical support, and Rayme Bio-technology, RWD Life Science Co., Ltd, and Revvity for equipment support.

## Author contributions

C.L., X.B., Z.Y., B.K., W.J., and W.P. conceived the project, designed the experiments, analyzed the data, and wrote the manuscript. L.Y., D.J., A.G., Y.M., S.W., L.W., X.W., Z.T., S.D., and K.T. conducted the experiments and analyzed the data. All authors contributed to the writing of the manuscript.

## Competing interests

The authors declare no competing interests.
