## [Peer Review File · Nature Communications]

REVIEWER COMMENTS

Reviewer #1 (Remarks to the Author):

This author proposed a molecular library based on vinpocetine-derived ionizable-lipid and developed a self-enhanced brain-targeted nucleic acid delivery system. Different from previous active brain targeting strategies, this article presented a novel conception, where enhanced brain permeability was achieved by utilizing the blood flow regulating function of vinpocetine. This is kind of interesting, but there still remain some concerns. I recommend publication of this manuscript after addressing the following issues:

1. The major concern is that, despite the enhanced brain targeting efficacy was observed in VIP nanoparticles, the existing evidence is insufficient to prove that the enhanced brain accumulation was achieved by blood flow regulating function of vinpocetine.
2. The representative ionizable lipid was A5-B1-C4.2 was selected and generated for further studied, why in Supplementary Table 2, the physiochemical characterizations of VIP were based on A1-B1-C2.1?
3. In page 10, line 202-203, the author claimed that “decreased uptake of VIP in the cell model with lower PDE1 expression suggested that VIP had protein-mediated targeting ability”. This is kind of confusing. Since the PDE-1 inhibition was reported to cause cerebral artery dilatation and thereby increasing cerebral blood flow in vivo, which, as claimed by author, could be beneficial to brain targeting. However, when in the in-vitro model, how could this achieved when “the blood flow” did even exist at all.
4. All the scale bar should be explained in the legend, please check and revise them.
5. It was puzzling why VIP could increase the in vitro BBB penetration efficacy? Is it related to the g-pg inhibiting function exerted by VIP? It is suggested that the control group should be set up accordingly to better illustrated its potential mechanism of BBB penetration. It seems that there was no rational basis for the selection of the control group. No convincing conclusions can be obtained through the presenting results, various factors such as size, zeta potential and et al, could not be ruled out when it comes to increased BBB penetration.

6. In “In vitro and in vivo brain targeting of VIP” part, the author used NP as a control nanoparticle, however the difference between NP and VIP was vague and should be explained when it was first introduced.

7. In order to validate the brain targeting was a result of blood flow regulating of VIP, it is suggested that vasoconstrictors or PDE-1 inhibitors should be applied to counter the vasodilating effects of VIP, therefore reconfirming the significant role of blood flow regulation in brain targeting ability of VIP.

8. In figure 3i, it seemed that the improved cerebral blood flow function mediated by VIP in brain tumor model was not as evident as those in other two brain diseases model, please explained why?

9. As is put by the author, the VIP was remarkably effective at facilitating brain-targeted drug delivery in various brain disease models (including AD, brain tumor, Fungal meningitis), while the diagram (Scheme 1) only displayed its multidimensional synergistic function on AD therapy, which is far from comprehensive and needs further revision.

Reviewer #2 (Remarks to the Author):

While this is an interesting study and new platform technologies for drug delivery across the BBB are needed, there are some issues with the study that must be addressed.

1. Typically in the mouse model of Cryptococcal meningitis (or cryptococcosis), a known strain of Cn is either administered via the nares/trachea to mimic the natural route of infection that occurs in humans or Cn is inoculated via the tail/femoral vein. Cn is highly neuroinvasive, and once it crosses the BBB it will proliferate and thrive in the CNS.

However, in this study a very large inoculum of Cn (500,000 cells) was delivered directly into the right side brain of the brain. This is likely to cause inflammation, astrocyte activation, gliosis etc., resulting in a leaky BBB. Without a measurement/assessment of the BBB to determine the extent of dysfunction/leakiness it is difficult to determine how effective VIP-AmB is.

Also, what is the half-life of VIP? Was the encapsulation efficiency or release of AmB determined?

2. There is no indication in this study whether mice inoculated with Cn were treated with AmB alone or with VIP alone.

Is VIP toxic to Cn? This needs to be determined.

3. In this study, mice were inoculated with 500,000 cells of *Cryptococcus neoformans* (Cn) expressing a luciferase gene in the right side of the brain. The type of strain used in this study was not provided thus it is not clear whether the strain used is a hypervirulent or attenuated for virulence. Was this a lab strain or a human isolate?

Was this the only inoculum size tested? Why was this particular size of inoculum chosen?

4. Standardized susceptibility testing is needed in order to assess whether this particular Cn strain used in the study is susceptible to amphotericin B, in other words, to confirm that the Cn strain is not resistant.

5. 48h after inoculation with Cn, mice were intravenously injected with LNP@AmB at a dose of 1mg/kg AmB (amphotericin B) every 5 days.

How was this dose determined?

6. AmB does not cross the BBB and for this reason, the standard of care includes flucytosine since it penetrates the BBB 60% - 95%. Given that AmB does not cross the BBB, the authors of this study loaded AmB into the VIP and determined used the model of fungal meningitis as a way to assess whether the VIP promoted crossing of AmB resulting in reduced fungal burden.

A positive control, like treatment with flucytosine to show clearance/resolution of Cn fungal burden would have strengthened the data.

7. The quality of the histology of Cn in the brain is poor. While lesions can be seen, cryptococci are not visible.

8. There is controversy in the literature about vinpocetine. Although the FDA has approved it (and many are critical of this), robust and rigorous data assessing its safety, pharmacokinetic/pharmacodynamic profile is lacking.

Reviewer #3 (Remarks to the Author):

The work entitled " Regulation of cerebral blood flow boosts brain penetration of vinpocetine-derived ionizable-lipidoid nanoparticles and their accumulation in lesions" from Bian et al. is interesting given that the BBB penetration capacity is one of the challenges in the different therapeutic approaches (pharmacological and non-pharmacological) for CNS diseases.

The use of nanoparticles that can cross the BBB and release nucleic acids as siRNAs targeting brain diseases is an approach that is not new, but its optimization and preclinical development is interesting to give us options for delivery and distribution in the CNS and all concerns about safety. In this work, the researchers propose a nanoparticle formed by vinpocetine (a natural product used for the treatment of different pathologies and that increases cerebral blood flow by inhibiting PDE1), called VIP. In addition, vinpocetine has antioxidant and anti-inflammatory activity that can be added to the cargo activity included in the nanoparticle.

This reviewer has focused on the chapter on in vivo experiment of Alzheimer's disease mice model and also on those of meningitis and glioblastoma, which, although much less developed, is of interest due to the possible potential of use of VIPs in various types of CNS disorders.

There are several points that have caught my attention, from different aspects of the work.

Methodologically, it should be clearly indicated how many days of treatment (1 injection every two days) the therapy is administered. From the scheme in Figure 4a, appears to be 14 days, but it should be indicated. Another important point is to indicate if during the days of the behaviour tests (10 more days) and until the euthanasia the VIP is also injected.

Regarding the treatment groups, a WT+PBS has been included but not a WT with a nanoparticle (as did for AD mice) to determine unwanted effects in a non-diseased animal.

This reviewer finds it very strange that the MWM (a test that stresses mice a lot) was performed first and after the NORT and not the other way around (it is usually the usual procedure first the least stressful tests and the last the most stressful), I would like the authors to give some reason.

In reference to the MWM, it would be desirable to show the 5-day learning curve in order to validate that the animals learn (in fact 5 days is a very short time) and therefore that on the day of the test they are all in the same conditions (except for the treatment obviously).

In both tests, VIP@scrambled also has a beneficial effect (in some cases similar to VIP@siBACE. This point is not disputed in deep and is highly significant.

This point is linked to the fact that the reduction of BACE levels is shown in animals treated with VIP@siBACE1 and also in scrambled, so my question is: could the positive effect in this AD model be

due to the action of vinpocetine, which the authors themselves mention has beneficial effects on CNS? The reduction in BACE1 gene expression should be quantified and correlated, together with the protein levels (Pierce's correlation) with the cognitive and histopathological improvement (BACE, pTAU activation, amyloid, etc.) shown by the authors in Figure 5 and other (pTAU parameters in the supplementary9), to demonstrate that the main effect is the silencing of this secretase and not the modulation of other pathways linked to neurodegeneration.

This point also applies to the meningitis model (amphotericin and inflammation) or glioma.

For example, GSK3 inhibition seems to be attributed to vinpocetine, and GSK3 inhibitors are being studied for AD at the clinical level, so it should be discerned without a shadow of a doubt that the beneficial effect is treatment with VIP@siBACE or it is additive effects and thus discussed. In fact, it seems to be more of the latter hypothesis that is being demonstrated in the AD model, given that both siBACE and scrambled nanoparticles show beneficial effect and a reduction in BACE protein levels.

Another important issue is that it is not shown whether vinpocetine improves blood flow in these animals, this point would also directly impact the beneficial action of injecting VIP, it should be shown if as in the control animals of the part in this model of AD; which also has flow compromised by vascular amyloidosis, vinpocetine may have a beneficial effect that could be correlated with its administration beyond silencing BACE. This would explain why VIP@scrambled has beneficial effects on almost all the parameters evaluated.

All these problems detected are timidly indicated in lines 325 to 327, but it requires more explanation and more data, given that the authors' hypothesis should vary substantially in terms of the therapeutic efficacy of their proposal.

In reference to the antioxidant and anti-inflammatory activity of vinpocetine, the phrase in 303 should be rephrased, including the involvement of inflammation (not just oxidative stress) in the genesis and progression of Alzheimer's disease

Minor: It should be explained why females are used and not both sexes (the sex should be indicated in the main document)

In figure g what is GSH@Amp? Nanoparticle of MC3 figure g and h has also been used and explained in text but not for GSH

In the methodology appeared LNP@sFAM but not siBACE at different points. In the methodology, the PEGASOS method of transparent tissues appears, but there is no data with this technique.

Bian X et al, Regulation of cerebral blood flow boosts brain penetration of vinpocetine-derived ionizable-lipidoid nanoparticles and their accumulation in lesions (NCOMMS-23-51861)

Point-by-Point Response to Reviewer Comments

We sincerely thank the editor and reviewers for their valuable feedback to improve the quality of our manuscript. The reviewers' comments are laid out below, followed by our responses. We have made extensive modifications to our manuscript and supplementary materials to address the comments and, we hope, to make our results more convincing. Changes to the revised manuscript are highlighted.

REVIEWER #1 (Remarks to the Author)

1. The major concern is that, despite the enhanced brain targeting efficacy was observed in VIP nanoparticles, the existing evidence is insufficient to prove that the enhanced brain accumulation was achieved by blood flow regulating function of vinpocetine.

Response: We sincerely appreciate this valuable comment, and we have added experiments to show that the enhanced brain accumulation was indeed achieved by the ability of VIP to regulate blood flow.

(1) We chose four agents known to reduce cerebral blood flow: the vasoconstrictor 5-hydroxytryptamine; the $\alpha 2$ receptor agonist dexmedetomidine; the NO synthase inhibitor *N*^G-monomethyl-L-arginine; and the $\alpha 1$ receptor agonist noradrenaline. We pretreated mice with each of these agents followed by injection of VIP-loaded fluorescent dye and found showed that each of these agents significantly reduced cerebral blood flow (**Supplementary Fig. S10 and Table S6**), indicating that regulation of cerebral blood flow enhanced the brain targeting of VIP. The corresponding experimental methods are described in the Supplementary Materials.

(2) We also measured cerebral blood flow in mouse models of Alzheimer disease, brain meningitis, and brain tumor at different stages by laser speckle flowgraphy. The results (**Supplementary Figure 11**) showed that the cerebral microvascular lesions and cerebral ischemia gradually increased as the disease progressed, and VIP improved cerebral blood flow in all models at every stage to achieve brain targeting.

2. The representative ionizable lipid was A5-B1-C4.2 was selected and generated for further studied, why in Supplementary Table 2, the physicochemical characterizations of VIP were based on A1-B1-C2.1?

Response: We apologize for our lack of clarity in describing these points.

(1) A1-B1-C2.1 was used as an initial structure to obtain a basic composition of VIP for subsequent screening of more diverse ionizable lipids. The physicochemical evaluation of VIP based on A1-B1-C2.1 (**Supplementary Table 2**) was used to determine the basic composition.

(2) After this basic composition, A5-B1-C4.2 was chosen through multiple rounds of screening. Details of the physicochemical characterizations of VIP based on A5-B1-C4.2 are shown in **Figure 2a-b** and **Supplementary Figures S3-S5**.

3. In page 10, line 202-203, the author claimed that “decreased uptake of VIP in the cell model with lower PDE1 expression suggested that VIP had protein-mediated targeting ability”. This is kind of confusing. Since the PDE-1 inhibition was reported to cause cerebral artery dilatation and thereby increasing cerebral blood flow in vivo, which,

as claimed by author, could be beneficial to brain targeting. However, when in the in-vitro model, how could this achieved when “the blood flow” did even exist at all.

Response: We appreciate the opportunity to respond to this valuable comment.

(1) PDE1 is secreted by cells near the cell membrane (*Pharmacol Rev* 2006;58:488–520; *Science* 2017;356: eaal3321), thereby allowing VIP to interact with PDE1 adjacent to the cell surface. Similarly, type IV collagenases, also secreted to the cell surface by tumor cells, have been shown to interact with CTT, a type IV collagenase inhibitor, and promote the association of liposomes with tumor cells *in vitro* (*Cancer Res* 2001; 61(10):3978-3985). For clarity, we changed the phrase “protein-mediated targeting ability” to “VIP may have been interacting with the PDE1 adjacent to the cell surface” in the revised manuscript (p 12, line 199).

(2) Because vinpocetine can inhibit p-gp and reverse drug efflux, we confirmed that VIP had a similar function, which contributed to the accumulation of fluorescent dye in the cells (**Supplementary Fig. S7a**). The corresponding methods are described in the Supplementary Materials (“Measurement of the cellular accumulation of rhodamine 123”).

4. All the scale bar should be explained in the legend, please check and revise them.

Response: With our thanks, we have double-checked and completed all of the scale bars, including those for **Figure 2(b), Figure 2(e), Figure 3(d), Figure 5(d), Figure 5(h), Figure 6(g-i), Supplementary Figure S15, and Supplementary Figure S20**.

5. It was puzzling why VIP could increase the in vitro BBB penetration efficacy? Is it related to the g-pg inhibiting function exerted by VIP? It is suggested that the control group should be set up accordingly to better illustrated its potential mechanism of BBB penetration. It seems that there was no rational basis for the selection of the control group. No convincing conclusions can be obtained through the presenting results, various factors such as size, zeta potential and et al, could not be ruled out when it comes to increased BBB penetration.

Response: We sincerely appreciate the opportunity to respond to these valuable comments.

(1) Regarding the choice of control group, we used NP as a negative control and MC3 as a commercially available LNP for comparison; the formulations are described in **Supplementary Table S5**. Moreover, particle sizes were similar for NP and VIP, and the particle size and zeta potential of MC3 were similar to those of VIP. Thus size and zeta potential could be ruled out as contributing to the regulation of BBB penetration.

(2) We appreciate the valuable suggestion and agree that the increased *in vitro* BBB penetration of VIP is related to its p-gp inhibition function. In the revised **Supplementary Figure S7a**, we showed that VIP significantly increased the intracellular accumulation of Rhodamine 123, a known p-gp substrate, similar to vinpocetine. We also updated the *in vitro* BBB assay to verify the permeability of VIP, with verapamil (a P-gp inhibitor) serving as a positive control. As shown in **Supplementary Figure S7b**, VIP increased the permeability of fluorescent dye to an extent similar to that of verapamil.

6. In “In vitro and in vivo brain targeting of VIP” part, the author used NP as a control nanoparticle, however the difference between NP and VIP was vague and should be explained when it was first introduced.

Response: We have added Supplementary Table S5 to illustrate the contrast between NP and VIP as recommended. We also explained the differences between NP and VIP at first mention (p 9, lines 164-165).

7. In order to validate the brain targeting was a result of blood flow regulating of VIP, it is suggested that vasoconstrictors or PDE-1 inhibitors should be applied to counter the vasodilating effects of VIP, therefore reconfirming the significant role of blood flow regulation in brain targeting ability of VIP.

Response: Thank you for this excellent suggestion; we have added experiments as described below.

(1) We chose four agents known to reduce cerebral blood flow through different mechanisms: the vasoconstrictor 5-hydroxytryptamine; the α_2 receptor agonist dexmedetomidine; the NO synthase inhibitor N^G -monomethyl-L-arginine; and the α_1 receptor agonist noradrenaline. All of these agents significantly reduced cerebral blood flow, as described in **Supplementary Table S6**.

(2) Pretreatment with these agents before injection of VIP@DiD led to significantly reduced accumulation of VIP-loaded fluorescent dye in the brain tissues (**Supplementary Figure S10**). This indicated that regulation of cerebral blood flow has a predominant role in brain targeting of VIP. The corresponding experimental methods are described in the revised Supplementary Materials (“Regulation of cerebral blood flow”).

8. In figure 3i, it seemed that the improved cerebral blood flow function mediated by VIP in brain tumor model was not as evident as those in other two brain diseases model, please explained why?

Response: We appreciate this important comment. We conducted additional experiments in three models of brain disease (brain tumors, meningitis, Alzheimer disease) that show differences in disease progression.

(1) In the original version of **Figure 3(i)**, the representative laser speckle flowgraphy image for the brain tumor model was taken late in the disease progression, when the brain lesions and ischemia were severe, obscuring visualization of the distribution of cerebral microvessels and hence obscuring any improvement in cerebral blood flow function mediated by VIP in that model relative to the others.

(2) To clarify this issue, we added more experiments to monitor cerebral blood flow in the brain tumor model at different stages without drug intervention (**Supplementary Figure S11c**). Again, as the brain tumor progressed, cerebrovascular lesions and ischemia became increasingly severe, and the cerebral microvascular distribution was difficult to observe. Nevertheless, VIP was found to improve cerebral blood flow after a single injection regardless of the disease stage. This highlights the significance of improving cerebral blood flow for delivering drugs to the lesion site and hence the potential of VIP-based multidimensional therapeutic strategies.

(3) Similar trends were observed in the meningitis and Alzheimer disease models, with cerebral microangiopathy and cerebral ischemia worsening over time. Again, VIP was shown to improve cerebral blood flow after a single injection in both models regardless of stage (**Supplementary Figure S11 a-b**).

9. As is put by the author, the VIP was remarkably effective at facilitating brain-targeted drug delivery in various brain disease models (including AD, brain tumor, Fungal meningitis), while the diagram (Scheme 1) only displayed its multidimensional synergistic function on AD therapy, which is far from comprehensive and needs further revision.

Response: As suggested, we have added more details to increase the comprehensiveness of the diagram (**Scheme 1** in the revised manuscript).

REVIEWER #2 (Remarks to the Author)

1. Typically in the mouse model of Cryptococcal meningitis (or cryptococcosis), a known strain of Cn is either

administered via the nares/trachea to mimic the natural route of infection that occurs in humans or Cn is inoculated via the tail/femoral vein. Cn is highly neuroinvasive, and once it crosses the BBB it will proliferate and thrive in the CNS.

However, in this study a very large inoculum of Cn (500,000 cells) was delivered directly into the right-side brain of the brain. This is likely to cause inflammation, astrocyte activation, gliosis etc., resulting in a leaky BBB. Without a measurement/assessment of the BBB to determine the extent of dysfunction/leakiness it is difficult to determine how effective VIP-AmB is.

Also, what is the half-life of VIP? Was the encapsulation efficiency or release of AmB determined?

Response: We sincerely appreciate these valuable comments, and we have added experiments to address these points as described below.

(1) We supplemented the experiment on BBB integrity by using Evans blue (EB) dye extravasation (*Sci Adv* 2024;10(2):eadj4260; *Curr Protoc Immunol* 2019;126(1):e83) as shown in **Supplementary Figure S18**, with the corresponding methods provided in the Supplementary Materials (“Evans blue tests”). The results showed good integrity of the BBB at the beginning of treatment (day 3 after brain infection).

(2) We also added a pharmacokinetics experiment and determined that the half-lives of compounds A5-B1-C4.2 and VIP were 7.26 ± 1.53 h and 14.10 ± 1.55 h, respectively (**Supplementary Figure S8**). Corresponding methods are described in the Supplementary Materials (“Pharmacokinetic Studies”).

(3) We also determined the encapsulation efficiency and *in vitro* cumulative release profiles of AMB in different formulations; the results are shown in the Supplementary Materials (**Supplementary Table S8** and **Supplementary Figure S17**).

2. There is no indication in this study whether mice inoculated with Cn were treated with AmB alone or with VIP alone. Is VIP toxic to Cn? This needs to be determined.

Response: Thank you for this valuable comment. We added two experiments to address these points.

(1) The commercially available AmBisome formulations was used as the 'AMB alone' treatment because of the poor water solubility of AMB. We also tested VIP alone *in vitro* and *in vivo* (**Supplementary Table S9** and **Figure S6e**). Corresponding methods are described in the Supplementary Materials (“Minimum inhibitory concentration assay”).

(2) Results of the MIC assay showed that VIP is nontoxic to Cn (MIC > 1024 µg/mL) (**Supplementary Table S9**).

3. In this study, mice were inoculated with 500,000 cells of *Cryptococcus neoformans* (Cn) expressing a luciferase gene in the right side of the brain. The type of strain used in this study was not provided thus it is not clear whether the strain used is a hypervirulent or attenuated for virulence. Was this a lab strain or a human isolate?

Was this the only inoculum size tested? Why was this particular size of inoculum chosen?

Response: We appreciate the opportunity to respond to these important points, and provided additional experiments as described below.

(1) The *Cryptococcus neoformans* (Cn) expressing a luciferase gene used in this study was a lab strain that had been transformed from the wild-type H99 strain, as described in the Supplementary Materials (“Strains”) (p 3, line 65). We also added an *in vivo* pharmacodynamics experiment with the wild-type H99 strain. We found that survival among the mice infected with H99 in the untreated group (**Figure 6f**) was similar to that of mice infected with H99-Luc without treatment (**Supplementary Figure S19c**), suggesting that the two strains have

comparable virulence.

In the previous tissue burden experiment, we injected 5 μ L of *C. neoformans* (500,000 cfu/mL) into the mouse brain; this concentration represents about 2,500 cfu/mouse. We chose this size for the inoculum after our preliminary experiments and according to several published articles (*Antimicrob Agents Chemother* 2018;62(11):e01315-18; *Med Mycol* 2016;54(3):280-286; *Antimicrob Agents Chemother* 2013;57(2):745-750; *J Antimicrob Chemother* 2007;60:162-165). In those studies, the inoculum size was 1,000–3,500 cfu/mouse, injected into the brain in a volume of 50–60 μ L.

4. Standardized susceptibility testing is needed in order to assess whether this particular Cn strain used in the study is susceptible to amphotericin B, in other words, to confirm that the Cn strain is not resistant.

Response: Thank you for this excellent suggestion. We have added standardized susceptibility testing to confirm that the Cn strain was susceptible to amphotericin B and was not resistant. The results are shown in **Supplementary Table 9**.

5. 48h after inoculation with Cn, mice were intravenously injected with LNP@AmB at a dose of 1mg/kg AmB (amphotericin B) every 5 days. How was this dose determined?

Response: Thank you for the opportunity to respond to this point.

We chose our dosing regimen (1 mg/kg of amphotericin B per day for 5 days) based on published references indicating that daily maintenance doses of amphotericin B is typically set at 0.5–1 mg/kg (*Am J Hosp Pharm* 1992;49(5):1156-1164), given once a day for 5 days (*Clin Infect Dis* 1999;28:42–8), once a day for 7 days (*AIDS* 1993;7:1018–9), or five doses given over 10 days (*Clin Infect Dis* 1996;22:938–43).

(1) We also conducted our own preliminary experiments on dose and frequency and found a daily dose of 1 mg/kg amphotericin B for 5 days to be both safe and effective.

6. AmB does not cross the BBB and for this reason, the standard of care includes flucytosine since it penetrates the BBB 60% - 95%. Given that AmB does not cross the BBB, the authors of this study loaded AmB into the VIP and determined used the model of fungal meningitis as a way to assess whether the VIP promoted crossing of AmB resulting in reduced fungal burden.

A positive control, like treatment with flucytosine to show clearance/resolution of Cn fungal burden would have strengthened the data.

Response: We sincerely appreciate this valuable comment.

We repeated the *in vivo* pharmacodynamic experiment and added a flucytosine treatment group as another positive control. Our results indicated that the flucytosine reduced fungal burden and prolonged the survival of the mice, whereas VIP had a better antifungal effect on meningitis than flucytosine (**Figure 6e** and **Supplementary Figure S20**).

7. The quality of the histology of Cn in the brain is poor. While lesions can be seen, cryptococci are not visible.

Response: Thank you for raising this important point. We repeated this experiment and found clear staining of *C. neoformans* (**Supplementary Figure S20b**).

8. There is controversy in the literature about vinpocetine. Although the FDA has approved it (and many are critical of

this), robust and rigorous data assessing its safety, pharmacokinetic/pharmacodynamic profile is lacking.

Response: Thank you for the opportunity to respond to this important issue.

Indeed, some published evidence suggests that vinpocetine has potential risks for pregnant women (*Reactions Weekly* 2019;1757:8). Abuse of vinpocetine may also result in adverse reactions when used in combination therapies (*Nat Prod Commun* 2016;11:607–609.). We added an experiment on the pharmacokinetics of the vinpocetine derivatives A5-B1-C4.2 and VIP (**Supplementary Figure S8**); updated the safety evaluation to include blood tests (**Supplementary Table S7**); and provided hematoxylin-and-eosin staining of major organs (**Supplementary Figure S15**), all of which indicated that the VIP used in our study was relatively safe at the given dosage.

REVIEWER #3 (Remarks to the Author)

There are several points that have caught my attention, from different aspects of the work.

Methodologically, it should be clearly indicated how many days of treatment (1 injection every two days) the therapy is administered. From the scheme in Figure 4a, appears to be 14 days, but it should be indicated. Another important point is to indicate if during the days of the behaviour tests (10 more days) and until the euthanasia the VIP is also injected.

Response: We sincerely appreciate these valuable comments, and we have modified our descriptions as suggested.

(1) We have re-written and clearly marked the dose regimen as “every 2 d for 2 weeks” in the manuscript (p 13, line 244) and in the Supplementary Materials (p 3, lines 76-80).

(2) We have re-written and clearly marked “At the end of the treatment” to show that the VIP treatment cycle ended before the behavioral tests took place (p 13, line 253 of the revised manuscript).

Regarding the treatment groups, a WT+PBS has been included but not a WT with a nanoparticle (as did for AD mice) to determine unwanted effects in a non-diseased animal.

Response: Thank you for the opportunity to respond to this important point. As suggested, we redid the in vivo pharmacodynamics experiment and the hematoxylin-eosin staining after adding the WT+VIP condition. Results showed that giving VIP every 2 days for 2 weeks had no harmful effects in non-diseased mice (**Supplementary Figure S15**).

This reviewer finds it very strange that the MWM (a test that stresses mice a lot) was performed first and after the NORT and not the other way around (it is usually the usual procedure first the least stressful tests and the last the most stressful), I would like the authors to give some reason.

In reference to the MWM, it would be desirable to show the 5-day learning curve in order to validate that the animals learn (in fact 5 days is a very short time) and therefore that on the day of the test they are all in the same conditions (except for the treatment obviously).

Response: Thank you for this excellent suggestion.

(1) The novel object recognition experiment was repeated directly after the 14-day dosing cycle (**Figure 4 h-i**). The results indicated that VIP@siBACE1-treated APP/PS1 mice exhibited a significant increase in NOR relative to the PBS-treated APP/PS1 control mice.

(2) We also added an experiment to demonstrate a 5-day learning curve for the MWM experiment (**Figure 4b**) in which we found that the APP/PS1 mice treated with VIP@siBACE1 showed decreased in escape latency

compared with the PBS group and showed a learning pattern similar to that of the WT group.

In both tests, VIP@scrambled also has a beneficial effect (in some cases similar to VIP@siBACE. This point is not disputed in deep and is highly significant.

This point is linked to the fact that the reduction of BACE levels is shown in animals treated with VIP@siBACE1 and also in scrambled, so my question is: could the positive effect in this AD model be due to the action of vinpocetine, which the authors themselves mention has beneficial effects on CNS? The reduction in BACE1 gene expression should be quantified and correlated, together with the protein levels (Pierce's correlation) with the cognitive and histopathological improvement (BACE, pTAU activation, amyloid, etc.) shown by the authors in Figure 5 and other (pTAU parameters in the supplementary9), to demonstrate that the main effect is the silencing of this secretase and not the modulation of other pathways linked to neurodegeneration.

This point also applies to the meningitis model (amphotericin and inflammation) or glioma.

For example, GSK3 inhibition seems to be attributed to vinpocetine, and GSK3 inhibitors are being studied for AD at the clinical level, so it should be discerned without a shadow of a doubt that the beneficial effect is treatment with VIP@siBACE or it is additive effects and thus discussed. In fact, it seems to be more of the latter hypothesis that is being demonstrated in the AD model, given that both siBACE and scrambled nanoparticles show beneficial effect and a reduction in BACE protein levels.

Response: We sincerely appreciate these valuable comments and kind suggestions.

- (1) At your suggestion, we added a qPCR experiment to quantify the reduction in BACE1 gene expression (**Supplementary Figure S13a**; methods given in Supplementary Materials ["Real-time PCR"]). We also added a Pierce correlation analysis of BACE1 gene expression levels with the cognitive and histopathologic improvement (BACE, p-tau, p-GSK3 β , and A β) (**Supplementary Figure S13b-h**). The results showed that silencing BACE1 was the main effect, rather than modulation of other pathways linked to neurodegeneration.
- (2) Numerous studies have reported that vinpocetine regulates various pathological proteins (BACE, p-tau, p-GSK3 β and A β), oxidative stress, and inflammatory environments in Alzheimer disease models (*Free Radic Res* 2000;33:497–506; *Cochrane Database Syst Rev* 2003;CD003119; *CNS Drugs* 2017;31:759–776; *Physiol Behav* 2019;208:112571). Our experimental results showed that VIP has similar functions (**Figure 5**). Therefore, VIP could have additive effects beyond silencing BACE1. As requested, we have expanded the corresponding discussion in the revised manuscript (p18, lines 329-332).
- (3) Vinpocetine has been shown to have neuromodulating effects in the meningitis model (*Front Cell Neurosci* 2021;15:680858) and in the glioma model (*Neurochem Int* 2022;154:105281), and so VIP may have a similarly additive effect. For these reasons, we coined the term "multidimensional brain targeting" to describe VIP-based drug delivery strategies.

Another important issue is that it is not shown whether vinpocetine improves blood flow in these animals, this point would also directly impact the beneficial action of injecting VIP, it should be shown if as in the control animals of the part in this model of AD; which also has flow compromised by vascular amyloidosis, vinpocetine may have a beneficial effect that could be correlated with its administration beyond silencing BACE. This would explain why VIP@scrambled has beneficial effects on almost all the parameters evaluated.

All these problems detected are timidly indicated in lines 325 to 327, but it requires more explanation and more data, given that the authors' hypothesis should vary substantially in terms of the therapeutic efficacy of their proposal.

Response: We sincerely appreciate these comments.

(1) At the reviewer's suggestion, we added monitored cerebral blood flow at different stages of progression in the Alzheimer model. The results revealed that vascular amyloidosis compromised cerebral blood flow as the disease progressed, and a single injection of VIP had a significant enhancement effect on cerebral blood flow (**Supplementary Figure S11a**). This would explain why VIP@scrambled had beneficial effects.

(2) Similar trends were observed in cerebral blood flow being compromised by cerebral microangiopathy with disease progression in the models of brain meningitis and brain tumor (**Supplementary Figure 11 Sb-c**); nevertheless, regardless of the stage, a single injection of VIP led to improved cerebral blood flow in both models.

(3) At the reviewer's request, we repeated the Nissl staining experiment to assess neuronal damage (**Figure 5h**), as described in the revised manuscript (pp 17-18, lines 324-329).

In reference to the antioxidant and anti-inflammatory activity of vinpocetine, the phrase in 303 should be rephrased, including the involvement of inflammation (not just oxidative stress) in the genesis and progression of Alzheimer's disease.

Response: We have revised this statement as suggested (p 17, lines 318-320).

Minor: It should be explained why females are used and not both sexes (the sex should be indicated in the main document)

Response: Thank you for the opportunity to address this valuable comment.

(1) Drug delivery systems typically require gender balance across several animal models to ensure the universality of targeting and efficacy. Therefore, we used male mice for the Alzheimer model and female mice for the brain tumor and meningitis models to comprehensively evaluate the VIP system.

(2) Female mice have been used extensively for the brain tumor and meningitis models (*J Neuroinflammation* 2019;16:25; *Br J Cancer* 2020;123:1633–1643; *Nat Nanotechnol* 2021;16:820–829; *Nat Commun* 2020;11:1521; *PLoS Negl Trop Dis* 2023;17(1):e0011068; *Infect Immun* 2004;72(4):2229-2239). After reviewing a large number of references, we concluded that no significant gender bias was present in these models of brain infection and brain tumors (*Neuro Oncol* 2023;25(5):995-1005; *Lancet Reg Health West Pac* 2023;34:100715; *Food Chem Toxicol* 1986;24(2):105-111).

(3) We indicated the sex of the mice in the main text (p13, line 249; p18, lines 347-348 and line 372).

In figure g what is GSH@Amp? Nanoparticle of MC3 figure g and h has also been used and explained in text but not for GSH

Response: GSH@AMB refers to GSH-LNP-loaded AMB. Glutathione (GSH) was used as the target moiety, as a glutathione-modified liposomal formulation is currently undergoing clinical trials for brain-targeted drug delivery. We thus chose GSH@AMB as a positive control in the pharmacodynamics experiment against meningitis (**Figure 6 and Supplementary Table S5**). We used the MC3@eGFP mRNA group as a control in our initial investigations of the ability of VIP to deliver large nucleic acid molecules. We have revised this description as suggested (p 12, lines 222-224).

In the methodology appeared LNP@sFAM but not siBACE at different points. In the methodology, the PEGASOS method of transparent tissues appears, but there is no data with this technique.

Response: In the Methods ("Intracellular Transfection Behavior and Mechanism"), we describe using LNP@FAM-siRNA primarily for tracking the intracellular behavior of siRNA rather than the gene-silencing effect of siBACE1, so siBACE1 was not specified. We have re-written this section to address this issue (p 20, line 365). We also clarified that the 3D hyalinization microscopic imaging data of brain tissue (**Figure 6i**) was obtained with the PEGASOS method.

REVIEWERS' COMMENTS

Reviewer #1 (Remarks to the Author):

accept

Reviewer #2 (Remarks to the Author):

In this study the authors exploited the neuro-protective properties of vinpocetine by developing a self-enhanced brain targeted nucleic acid delivery system. The authors were very thorough in responding to the suggestions and concerns raised by this reviewer. As a result of the all the changes, the data is strengthened and the conclusions are supportive of the data presented. The data and statistical analysis are robust. There is an enormous amount of work that went into this study as evidenced by the large amount of supplementary data. The work will be of interest to a wide audience.

One issue that I found:

Please re-format Supplementary Table S7. The last column in the table is not legible.

Reviewer #3 (Remarks to the Author):

Dear sirs

With reference to the answers given by the authors on the comments made, I must say that most of them have been adequately answered, however, there are still some points that need to be answered or modified in the revised documents

The results on NORT should be presented first (in the text and in the figure) or clearly indicate in the text that they are dealing with other sets of experiments, since, as I indicated in my first review, it is not correct to perform the cognitive experiments in the order proposed by the authors

In the added latency time graph, statistical analysis should be incorporated that allows the authors to conclude that the animals of the different groups learn differentially.

Authors do not answer why SNC experiments on AD mice were not done in both sexes, apart from issues of bioavailability etc, the variability between genders of cognitive responses is also very relevant. Females should be included in the study or adequately justified.

The gene expression results should be included in the description of the results on BACE modulation (p. 16) and if possible in Figure 5 and not as supplementary material since they are important enough to support the hypothesis to be in the main manuscript

I have not been able to identify the response on page 7 about the modification of the discussion on BACE on page 18 (329-332, which corresponds to the footnote of figure 5. Nor did I identify the response about Nissl on page 17, 324-329, which is again the caption of Figure 5.

These results should be included not only in the figures and caption but also in the main text

Thanks for incorporating data on blood flow in the brain, the results and their discussion should be incorporated into the manuscript, not just in the supplementary information as it is an important point that supports the authors' hypothesis and is part of the title

In general, the authors have carried out experiments that were required but have not incorporated them in a correct and understandable way into the text or discussion

Bian X et al, Regulation of cerebral blood flow boosts brain penetration of vinpocetine-derived ionizable-lipidoid nanoparticles and their accumulation in lesions (NCOMMS-23-51861A)

Point-by-Point Response to Reviewer Comments

We sincerely thank the editor and reviewers for the valuable feedback to improve the quality of our manuscript. The reviewer's comments are laid out below, followed by our responses. We have made comprehensive modifications to our manuscript and supplementary materials to address the comments and, we hope, to make our results more convincing. Changes to the revised manuscript are highlighted in **green** and provided the screenshot for your reference.

REVIEWER #2 (Remarks to the Author)

1. Please re-format Supplementary Table S7. The last column in the table is not legible.

Response: We sincerely appreciate this valuable comment, and we have made adjustments to our manuscript.

REVIEWER #3 (Remarks to the Author)

1. The results on NORT should be presented first (in the text and in the figure) or clearly indicate in the text that they are dealing with other sets of experiments, since, as I indicated in my first review, it is not correct to perform the cognitive experiments in the order proposed by the authors

In the added latency time graph, statistical analysis should be incorporated that allows the authors to conclude that the animals of the different groups learn differentially.

Response: We sincerely appreciate this valuable comment, and we have made adjustments to our manuscript.

(1) Specifically, we have reorganized the presentation of results pertaining to the Novel Object Recognition Test (NORT), prioritizing its exposition within the text (p13, line 269-280), as well as in **Figure 4**.

(2) Statistical analysis has been incorporated into the latency time graph (**Figure 4e**) to underpin the conclusions drawn from the data.

The screenshot as follows:

269 the novel object recognition (NOR)³⁸ and then morris water maze (MWM)³⁹ tests.

270 In the NOR test (Fig. 4b), after treatment with VIP@siBACE1, APP/PS1 mice showed a significant
 271 increase in NOR compared with PBS-treated APP/PS1 control mice, and the discrimination index (DI) and
 272 preference index (PI) for the novel object reached the performance levels of normal WT mice (Fig. 4c, d).

273 Results on the MWM test showed a similar trend: APP/PS1 mice treated with VIP@siBACE1 demonstrated a
 274 significant decrease in escape latency during the first 5 days of training relative to the PBS group and revealed
 275 a learning pattern similar to that of the WT group (Fig. 4e). In contrast, mice given PBS and MC3@siBACE1
 276 showed an aimless search strategy with no improvement in spatial learning or memory (representative tracking
 277 plots are shown in Fig. 4f), and no difference was noted in swimming speed (Fig. 4g). VIP@siBACE1-treated
 278 APP/PS1 mice showed the greatest improvement, with a 2.19-fold increase in the percentage of time spent in
 279 the target quadrant and the largest number of crossings compared with PBS-treated APP/PS1 control mice
 280 (Fig. 4h, i). These data confirm that VIP@siBACE1 mediates highly effective siRNA delivery to significantly

283 **Fig. 4. Behavioral evaluation of VIP@siBACE1 therapy in APP/PS1 mice.** (a) Schematic of the
284 experimental timeline; APP/PS1 and wild-type (WT) mice were treated with LNP@siBACE1 or PBS via tail
285 vein injection every 2 d (7 cycles). Mice were then subjected to the novel object recognition (NOR) and Morris
286 water maze (MWM) tests to evaluate memory, and samples were collected for molecular pathological
287 assessments. (Created with BioRender.com). (b) Setup for the NOR test. (c and d) Results from the NOR test.
288 (c) Discrimination index and (d) preference index of each group after LNP@siBACE1 treatment. (e–i) Data
289 in the MWM. (e) The 5-day learning curve for the MWM experiment. (f) Representative swimming track, (g)
290 swimming speed, (h) ratio of time spent in the target quadrant, and (i) numbers of crossing the platform
291 location of each group on the probe test day. Data are presented as means \pm SD (n=6 biologically independent
292 samples). * P <0.05, ** P <0.01, *** P <0.001, **** P <0.0001. Statistical significance was calculated with (c, e, h,
293 i) two-tailed unpaired t tests and (d) multiple t tests. Source data are provided as a Source Data file.

2. Authors do not answer why SNC experiments on AD mice model were not done in both sexes, apart from issues of bioavailability etc, the variability between genders of cognitive responses is also very relevant. Females should be included in the study or adequately justified.

Response: We appreciate the opportunity to address the insightful comment provided by the reviewer. We apologize for any misunderstanding regarding the previous query. The previous review 'Minor: It should be explained why females are used and not both sexes', focusing on the rationale behind the use of exclusively female subjects, we thought the inquiry pertained to the rationale for utilizing only female mice in the meningitis and brain tumor models. Consequently, our response primarily elucidated the rationale for these two models. Regrettably, our previous response did not adequately address the selection of male mice in the AD model. Please allow us to provide a detailed explanation.

(1) The primary objective of this study was to investigate the brain targeting capability of the VIP system by modulating cerebral blood flow in AD mice. A thorough literature review revealed that APP/PS1 mice exhibit structural damage and dysfunction in brain microvasculature (Microcirculation. 2015;22(2):133-145.; Neuroscience. 2018;385:246-254.; Annu Int Conf IEEE Eng Med Biol Soc. 2020;2020:3557-3560.). To assess whether the regulation of cerebral blood flow by VIP is influenced by sex, we have conducted preliminary experiment using female and male APP/PS1 mice. Our experimental findings align with the literature, confirming the presence of cerebral microvascular lesions in all APP/PS1 mice. Notably, both female and male mice exhibited a significant increase in cerebral blood flow following a single dose of VIP (**Supplementary Figure S12a**), indicating a consistent brain targeting mechanism in gender wherein VIP enhances cerebral blood flow (p13, lines 262-264).

(2) We chose to target BACE1 to examine whether systemic siRNA delivery by VIP could elicit pharmacological effects. Existing literature indicates that "No sex differences in BACE1 enzymatic activity were observed for any of the genotypes here studied." (Front Aging Neurosci. 2015;7:207.) ; "BACE1 levels are consistently reduced by ~50% in 5XFAD/BACE1+/- mice at all ages and are the same in both genders." and "females and males have the same level of BACE1." (Mol Neurodegener. 2015;10:1.); "male and female 5XFAD mice at 6-7 months of age showed equivalent levels of A β , BACE1, full-length APP and its metabolites." (Neuroscience. 2015;307:128-137.). Additionally, we explored whether there were any sex differences in the silencing effect of BACE1 protein in female and male APP/PS1 mice through preliminary

experiments. Our experimental findings are consistent with the literature. There is no discernible gender disparity in the expression of BACE1 protein in the cerebral cortex of APP/PS1 model mice, and VIP@siBACE1 exhibits a similar inhibitory effect on BACE1 protein in both female and male mice (p13, lines 262-264, **Supplementary Figure S12b**).

(3) Upon thorough review of numerous references, we have noted a prevailing trend of utilizing single-sex cohorts in studies involving APP/PS1 mice as models for Alzheimer's disease (AD) (**Male mice**: Pharmacol Rep. 2016;68(5):975-982.; Neural Plast. 2019;2019:2823679.; Behav Brain Res. 2019;376:112182.; Exp Neurol. 2019;311:67-79.; Clin Transl Med. 2020;10(3):e50.; J Neuroinflammation. 2020;17(1):258.; Food Chem Toxicol. 2021;151:112105.; **Female mice**: Biomed Res Int. 2016;2016:8380618.; EBioMedicine. 2019;45:393-407.; Neural Regen Res. 2021;16(10):2078-2085. Phytomedicine. 2022;100:154056.; J Neurosci. 2022;42(6):1154-1165.; Acta Pharmacol Sin. 2022;43(9):2226-2241.; Neurobiol Aging. 2024;135:60-69.). Additionally, the purpose of the majority of articles utilizing female mice is to underscore the impact of estrogen on dementia.

The screenshot as follows:

262 of siRNA could elicit pharmacologic effects. After preliminary experiments using female and male APP/PS1
263 mice, we determined that there was no discernible gender bias concerning cerebral blood flow enhancement
264 and BACE1 protein silencing in the APP/PS1 mice model treated by the VIP system (Supplementary Fig. S12).

230

231 **Supplementary Figure S12.** Gender bias of cerebral blood flow enhancement and BACE1

232 protein silencing in the APP/PS1 mice model treated by the VIP system. (a) Blood flow

233 enhancement effect of VIP on brain microvessels detected by using laser speckle flowgraphy in

234 the AD models. Data are presented as means \pm SD, were obtained by measuring the blood

235 perfusion unit every 10 minutes. Data are representative of two independent experiments with

236 similar results. (b) Western blot data for BACE1 protein expression in cortex from control

237 APP/PS1 groups, and VIP@siBACE1-treated APP/PS1 mice. Quantification of western blotting

238 analysis of BACE1 protein expression is shown relative β -actin. Data are presented as means \pm SD

239 ($n=3$ biologically independent samples). * $P<0.05$, NS means no significance. Statistical

240 significance was calculated with two-tailed unpaired t tests. Source data are provided as a Source

241 Data file.

3. The gene expression results should be included in the description of the results on BACE modulation (p. 16) and if possible in Figure 5 and not as supplementary material since they are important enough to support the hypothesis to be in the main manuscript

Response: We sincerely appreciate the valuable comments and thoughtful suggestions provided by the reviewer.

In response to your recommendation, we have incorporated the gene expression results into Figure 5 (**Figure 5f**) and expanded the corresponding explanation in the revised manuscript (p15, lines 314-319).

The screenshot as follows:

330 presented as means \pm SD (n=3 biologically independent samples). (f) BACE1 relative mRNA expression level
 331 in cortex was quantified by q-PCR. Data are presented as means \pm SD (n=3 biologically independent samples).

334 treated with VIP@siScr. Furthermore, the quantification of BACE1 gene expression showed that the BACE1
 335 mRNA expression in the cortex of APP/PS1 mice was significantly inhibited after VIP@siBACE1 treatment
 336 (Fig. 5f). Combined with the Pearson correlation analysis of BACE1 gene expression levels with the protein
 337 levels with the cognitive and histopathologic improvement (Supplementary Figs. S14), the results showed that
 338 VIP could have additive effects beyond inhibiting BACE1, but silencing BACE1 was the main effect, rather
 339 than modulation of other pathways linked to neurodegeneration.

4. I have not been able to identify the response on page 7 about the modification of the discussion on BACE on page 18 (329-332, which corresponds to the footnote of figure 5. Nor did I identify the response about Nissl on page 17, 324-329, which is again the caption of Figure 5.

These results should be included not only in the figures and caption but also in the main text
Response: We apologize for any confusion caused by the lack of clarity in demonstrating the modifications made to the discussion regarding BACE and the response regarding Nissl staining. To address this, we have highlighted the revised sections in green within the manuscript (pages 17, lines 349-356) and provided a screenshot for your reference.

The screenshot as follows:

349 phosphorylated tau levels. Therefore we used Nissl staining to assess neuronal damage in the brains of AD
350 mice, focusing on the CA1 and CA3 regions of the hippocampus (commonly associated with memory)⁴⁵. Nissl
351 staining revealed deeper nuclear staining and cell body shrinkage in the CA1 and CA3 regions of the
352 hippocampus in APP/PS1 mice treated with PBS and MC3@siBACE1, in contrast with findings in WT mice
353 and those treated with VIP@siRNA (Fig. 5). The results suggest that VIP@siRNA has neuroprotective effects.
354 In conclusion, our use of the term 'multidimensional synergistic brain-targeted drug delivery system' reflects
355 the ability of VIP to enact an integrated neuromodulatory function, inhibiting the GSK3 β /BACE-1 signaling
356 cascade in addition to its conventional antioxidant and anti-inflammatory effects.

5. Thanks for incorporating data on blood flow in the brain, the results and their discussion should be incorporated into the manuscript, not just in the supplementary information as it is an important point that supports the authors' hypothesis and is part of the title

Response: We deeply appreciate this insightful comment.

(1) In response to your suggestion, we have elaborated on the relevant discussion in the revised manuscript (p12, lines 248-254).

(2) **Figure 3i** in the manuscript corresponds to the representative data presented in the added **Supplementary Figure S11**. Given their correlation, the cerebral blood flow data for the three pathological models at different stages are included in the supplementary materials.

The screenshot as follows:

248 low cure rates. We verified experimentally that the three representative models all had cerebral microvascular
249 blockage and decreased local cerebral blood flow^{10, 12, 34}, which could restrict drug accumulation. As the
250 disease progressed, the cerebrovascular lesions and ischemia became increasingly severe, and the cerebral
251 microvascular distribution was difficult to observe. Nevertheless, VIP was found to improve cerebral blood
252 flow after a single injection regardless of the disease stage. In conclusion, we found that VIP improved cerebral
253 blood flow in normal mice and also enhanced cerebral blood flow at the lesion site in these three pathological
254 models at different stages (Fig. 3i and Supplementary Fig. S11), thereby enhancing brain targeting. Notably,